# DENR promotes translation reinitiation via ribosome recycling to drive expression of oncogenes including *ATF4*

Jonathan Bohlen [1,2,3,4,5], Liza Harbrecht[1,3], Saioa Blanco[1,3,5], Katharina Clemm von Hohenberg[1,2,3,5], Kai Fenzl [1,3,4,6], Günter Kramer [1,5,6], Bernd Bukau[1,5,6] & Aurelio A. Teleman [1,2,3,5 ✉]

Translation efficiency varies considerably between different mRNAs, thereby impacting protein expression. Translation of the stress response master-regulator ATF4 increases upon stress, but the molecular mechanisms are not well understood. We discover here that translation factors DENR, MCTS1 and eIF2D are required to induce ATF4 translation upon stress by promoting translation reinitiation in the ATF4 5′UTR. We find DENR and MCTS1 are only needed for reinitiation after upstream Open Reading Frames (uORFs) containing certain penultimate codons, perhaps because DENR•MCTS1 are needed to evict only certain tRNAs from post-termination 40S ribosomes. This provides a model for how DENR and MCTS1 promote translation reinitiation. Cancer cells, which are exposed to many stresses, require ATF4 for survival and proliferation. We find a strong correlation between DENR•MCTS1 expression and ATF4 activity across cancers. Furthermore, additional oncogenes including *a-Raf*, *c-Raf* and *Cdk4* have long uORFs and are translated in a DENR•MCTS1 dependent manner.

[1] German Cancer Research Center (DKFZ), 69120 Heidelberg, Germany. [2] CellNetworks—Cluster of Excellence, Heidelberg University, Heidelberg, Germany. [3] Heidelberg University, 69120 Heidelberg, Germany. [4] Heidelberg Biosciences International Graduate School (HBIGS), Heidelberg, Germany. [5] National Center for Tumor Diseases (NCT), partner site, Heidelberg, Germany. [6] Center for Molecular Biology of Heidelberg University (ZMBH), DKFZ-ZMBH Alliance, Im Neuenheimer Feld 282, Heidelberg D-69120, Germany. ✉email: a.teleman@dkfz.de

Cellular protein levels do not correlate well with mRNA levels in both healthy cells and cancer cells[1,2], thereby highlighting the importance of regulated mRNA translation in shaping cellular proteomes, and hence cellular phenotypes. Indeed, translation efficiency (TE) can vary considerably between different mRNAs[2–4]. Dysregulation of mRNA translation is frequently observed in cancer where it contributes to oncogenesis and cancer progression[5].

One regulatory element present in roughly 50% of human mRNAs is an upstream open reading frame (uORF)[6]. uORFs are often translated by ribosomes as they scan from the mRNA 5′ end[2], thereby necessitating either leaky scanning or translation reinitiation to translate the main downstream ORF[7–9]. This imparts regulatory potential to uORFs. In cases where ribosomes translate a uORF, they need to terminate translation before they can reinitiate on the main downstream ORF. Termination and recycling are multistep processes involving peptide release, 60S subunit dissociation, release of deacylated tRNAs and ultimately dissociation of the post termination 40S ribosome from the mRNA[10–13]. To reinitiate translation after a uORF, however, 40S ribosomes need to release the deacylated tRNA but remain associated with the mRNA, then recruit a new initiator tRNA, and resume scanning. These reinitiation processes are currently not well understood[7,8].

The best studied mRNA regulated by translation reinitiation in animals is Activating Transcription Factor 4 (ATF4), the main downstream effector of the integrated stress response (ISR)[14]. When exposed to stresses including proteotoxic stress, dsRNA, low amino acids, or heme deficiency, the kinases PERK, PKR, GCN2, or HRI, respectively, phosphorylate and inactivate eIF2α[14,15]. This shuts down global translation, while concomitantly activating translation of ATF4 via reinitiation. ATF4 then orchestrates the cell's response to stress. For instance, ATF4 is required and sufficient to trigger the cell's amino acid-responsive transcription program[16] including transcription of asparagine synthetase (ASNS)[14] and to induce autophagy[14]. ATF4 is also a developmental transcription factor required for proper formation of the hematopoietic system, brain, bone, and eye[17–21]. Finally, the ISR and ATF4 are mediators of tumor resistance to stress and therapy[22–28]. Human ATF4 has three uORFs in its 5′ UTR. The 3rd uORF (uORF3) overlaps with the start codon of the main ATF4 ORF, so its translation is mutually exclusive with that of ATF4 (Supplementary Fig. 1a). In the absence of stress, ribosomes are thought to translate uORF1 and 2 and then quickly recharge with initiator tRNA to reinitiate translation on uORF3. If eIF2α activity is low, ribosomes recharge with ternary complex or iMet-tRNA more slowly[29], scan past the uORF3 ATG, and instead reinitiate translation on the main ATF4 ORF[30–32]. Recent data suggest the mechanism may be more complex[33,34]. Although ATF4 is the canonical model mRNA for translation reinitiation in animals, the factors required to promote translation reinitiation on the ATF4 mRNA are not known.

The DENR•MCTS1 complex and eIF2D, which combines the domains of DENR and MCTS1 into a single protein, recycle post-termination 40S ribosomes in vitro and in yeast[35,36] and can stimulate eIF2-independent recruitment of initiator tRNA to the ribosome[37]. We previously reported that DENR•MCTS1 promotes translation reinitiation in Drosophila and human cells[9,38]. In human cells DENR is mainly required to translate mRNAs bearing very short 1-aa long uORFs consisting of a start codon followed directly by a stop codon (e.g. ATGTGA)[9,38]. These mRNAs are enriched for neuronally expressed genes, and fittingly, DENR mutations are found in autism patients where they cause neuronal migration defects[39]. Notably, however, these factors have also been linked to cancer: MCTS1 is amplified in T-cell lymphoma[40], DENR•MCTS1 transform cells and promote

tumor progression[41,42], and DENR expression correlates with poor cancer prognosis[43]. So far, however, none of these pro-cancer effects of DENR•MCTS1 have been linked to direct translational targets, suggesting additional targets of DENR•MCTS1 remain to be discovered. Here we find that DENR•MCTS1 and eIF2D are required for translation of ATF4 and other oncogenes such as a-Raf, c-Raf, and CDK4 by promoting translation reinitiation after uORFs in the 5′UTRs of these mRNAs. The dependence on DENR is due to the identity of the penultimate codon of these uORFs. This is likely because only certain tRNAs require DENR to be removed from post-termination 40S ribosomes.

## Results

**Ribosome footprinting identifies DENR-target mRNAs.** To identify mRNAs requiring DENR for efficient translation, we generated, using CRISPR/Cas9 and two different sgRNAs, two HeLa cell lines lacking DENR protein (DENR^KO1 and DENR^KO2) (Supplementary Fig. 1b, c). In agreement with previous results[38], loss of DENR causes reduced translation of a luciferase reporter bearing a synthetic 1-amino acid long uORF (Supplementary Fig. 1d), but not a luciferase reporter bearing a longer uORF (Supplementary Fig. 1e), confirming that DENR selectively supports reinitiation after certain uORFs. DENR^KO cells are viable but display a proliferation defect (see below). We performed ribosome footprinting[4] on the control and DENR^KO cell lines, yielding the expected quality controls (enrichment of footprints in the ORF compared to UTRs, triplet periodicity, and average footprint length, Fig. 1a, b and Supplementary Fig. 2a). Note that in all figures of this paper we assign footprint positions to their 5′ends, resulting in a 15 nt offset to features such as start and stop codons (Fig. 1a, b). Generally, the footprint densities in main ORFs look similar in the control and DENR^KO cells, except for an accumulation of 80S footprints 45 nt upstream of termination codons in DENR^KO cells (Fig. 1b). This was also observed in Tma22 (DENR) and Tma64 (eIF2D) double-mutant yeast, and it was ascribed to 80S ribosomes queuing in front of stalled post-termination 40S ribosomes[36], consistent with the known 40S recycling activity of DENR•MCTS1 in vitro[35]. A plot of observed Z-score vs. theoretical Z-score for fold change in mRNA TE ($\log_2$FC(TE)) upon DENR^KO identified 517 genes with significantly reduced translation and none with increased translation (Fig. 1c and Supplementary Data 1). Likewise, analysis using Xtail[44] identified mainly genes with significantly reduced TE in the DENR^KO (Supplementary Fig. 2b). These DENR-targets overlap significantly with the human orthologs of DENR targets previously reported in mouse cells[45] (Supplementary Fig. 2c). Down-translated genes were enriched for transcription factors and kinases (Supplementary Fig. 2d), including ATF4, a-Raf (ARAF), c-Raf (RAF1), and PI3K regulatory subunits (Fig. 1c). Amongst the most translationally downregulated mRNAs were mRNAs with 1-aa long uORFs such as Drosha and MAP2K6. These displayed increased ribosome density on the 1-aa uORFs (indicated as red boxes, Fig. 1d and Supplementary Fig. 2e) and reduced ribosome density on the main ORF, consistent with an inability of ribosomes to reinitiate on a main ORF downstream of a 1-aa uORF in DENR^KO cells. Indeed, transcriptome-wide, mRNAs with 1-aa uORFs had significantly reduced TE (Fig. 1e and Supplementary Fig. 2f). mRNAs with longer uORFs also had significantly reduced TE transcriptome-wide (Fig. 1e), and visual inspection confirmed that some of the top downregulated mRNAs such as a-Raf and c-Raf only contain longer uORFs (Supplementary Fig. 2g–i). This raised the possibility that

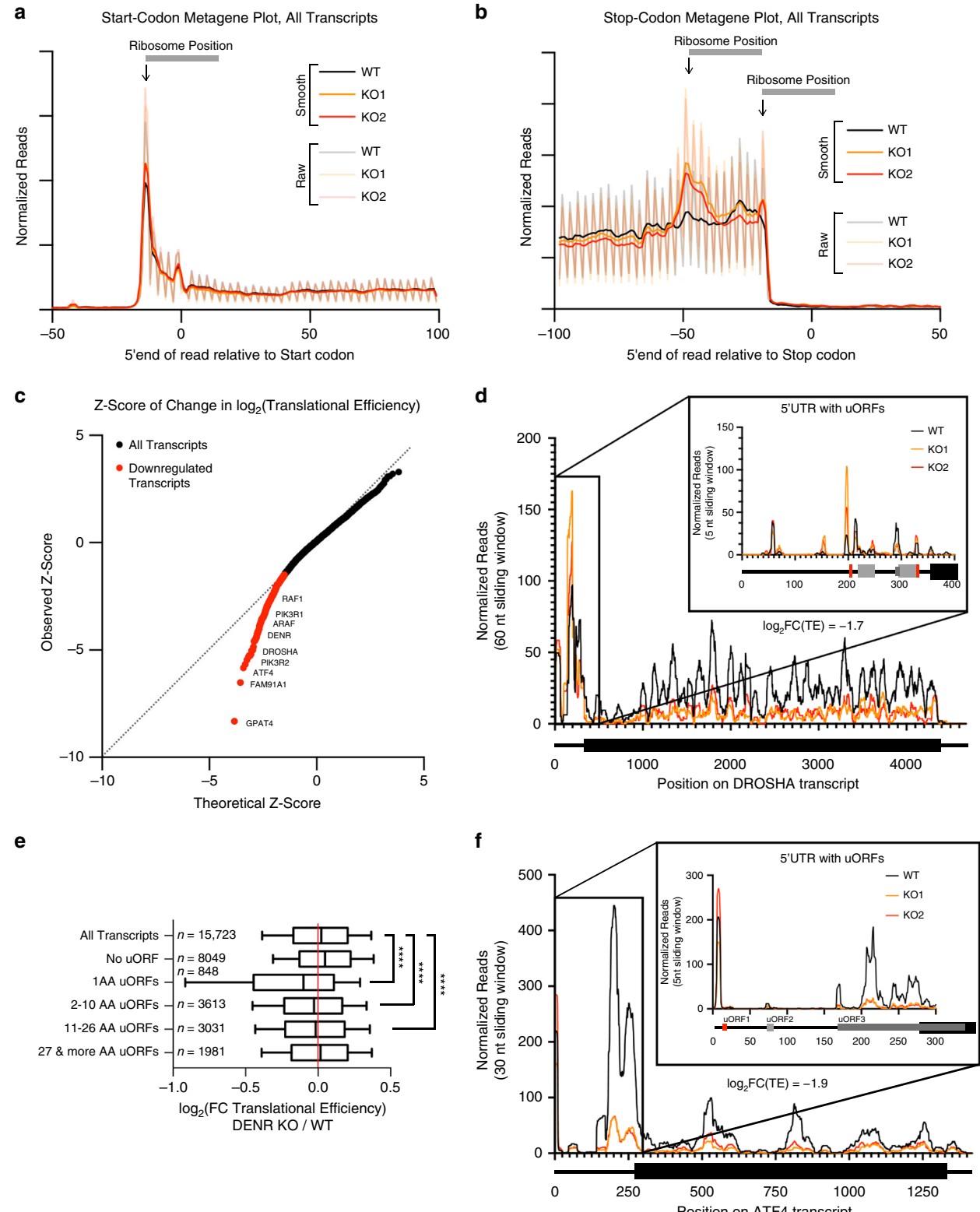

DENR•MCTS1 may also promote translation reinitiation downstream of a certain subset of longer uORFs. The third most translationally downregulated mRNA transcriptome-wide is *ATF4*, which contains both a 1-aa uORF (uORF1) as well as longer uORFs. *DENR*KO cells have unchanged footprint density on uORF1 and uORF2, but reduced footprint density on uORF3 as well as the main *ATF4* ORF (Fig. 1f), indicating DENR•MCTS1 are required for efficient translation of *ATF4*.

**DENR•MCTS1 are required for efficient ATF4 translation.** To study the involvement of DENR•MCTS1 in *ATF4* translation, we first assayed ATF4 protein levels in control and *DENR*KO cells (Fig. 2a). Control HeLa cells have low levels of ATF4, which increase in response to tunicamycin-induced endoplasmic reticulum (ER) stress (lanes 1–2, Fig. 2a). Upon tunicamycin treatment, eIF2α phosphorylation appears to increase only mildly (Fig. 2a, b), probably because p-eIF2α(Ser51) antibodies also

**Fig. 1 DENR promotes reinitiation after short and long uORFs. a**, **b** Metagene profiles for 80S ribosome footprinting of control and DENR knockout ("KO1", "KO2") HeLa cells showing the position of the 5′end of ribosome footprints relative to **a** start and **b** stop codons. Read counts were normalized to sequencing depth. "Smooth" indicates the curve was smoothened with a 3 nt sliding window. **c** Observed $Z$-score vs. theoretical $Z$-score plot for the $\log_2$(fold change in translation efficiency) in control versus DENR knockout HeLa cells of 7788 detected transcripts. 517 translationally downregulated transcripts (red). No upregulated transcripts were identified by this analysis. **d** Ribosome occupancy on the *DROSHA* transcript and 5′UTR (inset). Read counts were normalized to sequencing depth and scaled to mRNA abundance (one value for each cell line), graphs were smoothened with a sliding window of 60 nt (main plot) or 5 nt (inset). 5′UTR features: 1 amino acid uORFs (red), longer uORFs (gray). **e** Change in translational efficiency of groups of transcripts containing uORFs of indicated lengths. Transcripts in each group can also contain uORFs from other groups, except transcripts with 1 AA uORFs, which were excluded from all other groups. ****$p < 10^{-6}$ nonparametric Kruskal–Wallis test corrected for multiple comparisons. Box middle = median, hinges = quartiles, whiskers = deciles. **f** Ribosome occupancy on the *ATF4* transcript and 5′UTR (inset). Read counts were normalized to sequencing depth and scaled to mRNA abundance (one value for each cell line), and graphs were smoothened with a sliding window of 30 nt (main plot) or 5 nt (inset). 5′UTR features: 1 amino acid uORFs (red), longer uORFs (gray). Source data are provided as a Source Data file.

detect unphosphorylated eIF2α (Supplementary Fig. 3a), yielding a high background signal. Nonetheless tunicamycin efficiently activated the ISR, judged by a global inhibition in translation (Supplementary Fig. 3b) and clear induction of ATF4 (Fig. 2a and Supplementary Fig. 3a). *DENR*$^{KO}$ cells have strongly reduced ATF4 protein levels, both in the absence and in the presence of stress (lanes 3–6, Fig. 2a), but not reduced *ATF4* mRNA levels (Supplementary Fig. 3c), consistent with impaired ATF4 translation in these cells. Reduced ATF4 protein levels are also observed in *MCTS1*$^{KO}$ HeLa cells (lanes 7–9, Fig. 2a and Supplementary Fig. 3d). (Note that DENR and MCTS1 are mutually dependent on each other for protein stability, as previously reported[46]). The reduced ATF4 protein levels can be rescued by expressing DENR from a plasmid in the *DENR*$^{KO}$ cells (Fig. 2b, lane 8), confirming the phenotype is on-target. *DENR*$^{KO}$ cells still express eIF2D, which contains the functional domains of both DENR and MCTS1 combined into one protein. Knockdown of *eIF2D* in the *DENR*$^{KO}$ cells further reduced ATF4 protein levels and strongly impaired the stress-inducibility of ATF4 (Fig. 2c). Knockdown of *DENR* and/or *eIF2D* also reduced ATF4 protein levels in HT1080 fibrosarcoma cells (Supplementary Fig. 3e) indicating this is not specific to HeLa cells. The contribution of eIF2D to ATF4 translation varies between cell lines, perhaps due to differences in eIF2D activity or abundance.

To test whether DENR•MCTS1 act on ATF4 translation via its 5′UTR, we cloned the ATF4 5′UTR into a luciferase reporter (Fig. 2d). Indeed, *DENR*$^{KO}$ cells have reduced expression of the ATF4-luciferase reporter (~45%), both in the presence and absence of cell stress, and this can be rescued by re-expressing DENR from a plasmid (Fig. 2d). Likewise, expression of luciferase reporters containing the 5′UTRs of other top hits from the ribosome footprinting, including *a-Raf* and *c-Raf* which have long uORFs, are also DENR dependent (Fig. 2e, the *a-Raf* and *c-Raf* reporters drop 58 and 63% upon *DENR*$^{KO}$, respectively). Loss-of-function for eIF2D had an additive effect on some targets, indicating the two complexes have partially overlapping functions (Fig. 2f and Supplementary Fig. 3f). As for ATF4, protein levels of endogenous a-Raf and c-Raf were reduced upon DENR knockout or knockdown (Fig. 2g and Supplementary Fig. 3g, with antibody validation in Supplementary Fig. 3h), or upon double DENR/eIF2D depletion (Fig. 2c lanes 4 and 6) thereby identifying also a-Raf and c-Raf as DENR targets.

**DENR is required for reinitiation after certain uORFs.** To identify which elements in the 5′UTR of *ATF4* are responsible for imparting DENR dependence, we dissected the *ATF4*-luciferase reporter and assayed activity in the presence and absence of stress. Mutation of the start codons of uORF1 and 2 caused the *ATF4* reporter to lose DENR dependence (reporter #2, Fig. 3a). Translation of this reporter, however, is low both in the presence and absence of stress, because in the absence of uORF1 and 2

translation initiates on uORF3, and hence not on the overlapping main ORF[31]. We therefore also removed uORF3, causing the reporter to be expressed at high levels, yet it is still DENR independent (reporter #3, Fig. 3a). Together, these data suggest that DENR dependence of *ATF4* is imparted by uORF1 and/or 2. To determine whether one or both of the uORFs is responsible for DENR dependence, we tested each one in isolation. Indeed, the region of the *ATF4* 5′UTR containing the 1-aa long uORF1 imparts DENR dependence to the reporter (~60% drop in *DENR*$^{KO}$), and this is abolished if the ATG of uORF1 is mutated, or if uORF1 is extended by insertion of 3 TCC$^{Ser}$ codons in front of the stop codon to become 4 amino acids long (reporters #4–6, Fig. 3a). The region containing uORF2 (in which the ATG of uORF1 is mutated) also imparts DENR dependence (reporter #7, ~50% drop in *DENR*$^{KO}$) and this is abolished by mutating the ATG of uORF2 (reporter #3). In sum, both uORFs 1 and 2 contribute to the DENR dependence of ATF4 translation.

This was unexpected to us, given that the *ATF4* uORF2 is three amino acids long, and in our hands translation downstream of 3-aa uORFs is usually not DENR dependent (e.g. Supplementary Fig. 1e). Extension of uORF2 to six codons, by inserting 3 TCC codons behind the start codon, caused it to remain DENR dependent (reporter #8, Fig. 3a). Likewise, the 5′UTR of *a-Raf* also contains two long uORFs (Supplementary Fig. 2g). Mutation of the ATG of each uORF showed that both contribute to its DENR dependence (Fig. 3b). The 5′UTR of *c-Raf* also contains two long uORFs and a third 1-aa uORF nested at the end of the second uORF (Supplementary Fig. 2g). Deletion of the ATG of each of these three uORFs in various combinations revealed that the last two uORFs contribute to rendering translation of c-Raf DENR dependent (Supplementary Fig. 4a). In sum, several of the mRNAs that show highest dependence on DENR•MCTS1 for translation according to our ribosome footprinting are DENR dependent due to the presence of uORFs that are >1-aa long.

**Certain penultimate codons make uORFs DENR•MCTS1 dependent.** We next asked what features of these longer uORFs cause DENR dependence. Although mutation experiments revealed that the upstream and downstream regions are not important (not shown), the long uORF itself, including its Kozak context, is sufficient to impart DENR dependence if cloned into the 5′UTR of *Lamin B* (reporter #2, Fig. 3c). We noticed that the amino acid sequence of the *a-Raf* uORF is similar across various animals (Supplementary Fig. 4b), as is the amino acid sequence of *ATF4* uORF2 (Supplementary Fig. 4c) suggesting the uORF sequence may be functionally relevant. Indeed, mutating the whole amino acid sequence of *a-Raf* uORF1 to poly-TCC$^{Ser}$ abolished it is DENR dependence (reporter #3, Fig. 3c). Targeted codon mutations revealed that the two codons before the stop codon contribute to DENR dependence (Fig. 3c). Likewise, mutation of the *ATF4* uORF2 coding sequence caused expression

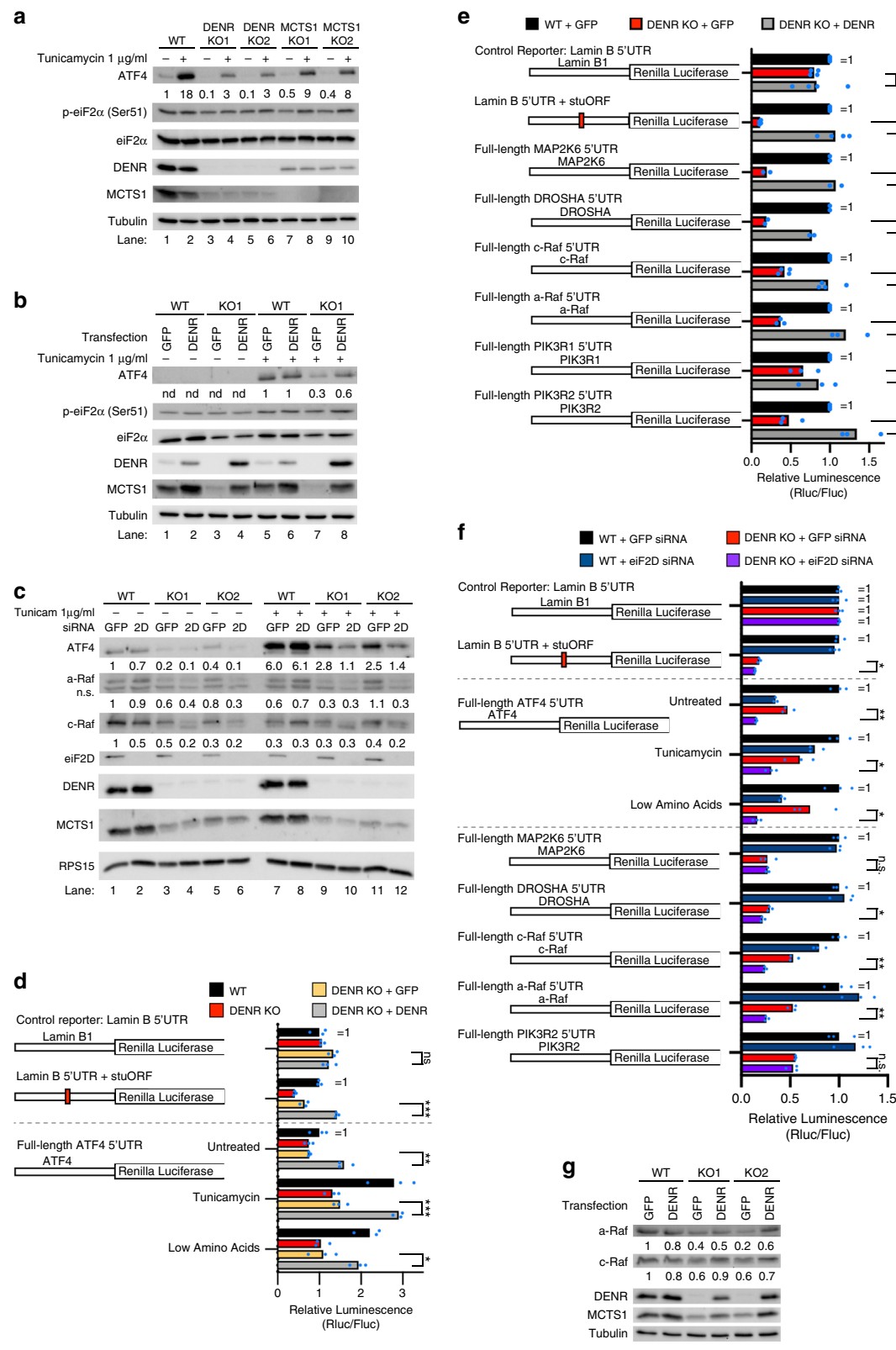

of the reporter to become DENR independent (Supplementary Fig. 5a). Interestingly, the penultimate (second to last) codon of *a-Raf* uORF1 and the third-to-last codon of *ATF4* uORF2 are both GCG^Ala, suggesting codon identity might be important. Therefore, we generated synthetic reporters containing 2-aa uORFs where the first codon is ATG and the second codon was varied. This confirmed that a GCG^Ala penultimate codon renders

the luciferase reporter ~40% DENR dependent (Fig. 3d). Testing of uORF terminal codons in a more comprehensive manner revealed a gradation of effects, with most codons causing no DENR dependence, a few codons such as GCG^Ala causing significant DENR dependence (Supplementary Fig. 5b), and some codons causing intermediate DENR dependence (CTG^Leu, 25% drop, Fig. 3d).

**Fig. 2 DENR•MCTS1 and eIF2D promote ATF4, c-Raf, and a-Raf translation. a** DENR and MCTS1 knockout cells have defective ATF4 induction upon ER-stress. Cells were treated with 1 µg/ml tunicamycin for 16 h. Results are representative of three biological replicates. **b** Defect in ATF4 induction is rescued by reconstitution with DENR. Control or *DENR*[KO] cells were transfected with either GFP or DENR expression plasmid, reseeded and treated with 1 µg/ml tunicamycin for 16 h. Results are representative of three biological replicates. **c** Cells lacking DENR and eIF2D show little induction of ATF4 upon stress. Control or *DENR*[KO] HeLa cells were transfected with siRNA targeting either GFP or eIF2D mRNA. Western blot showing reduced ATF4, a-Raf, and c-Raf protein levels. Results are representative of three biological replicates. **d** Translation of the *ATF4* 5′UTR reporter is DENR dependent both during unstressed and stress conditions. Treatment for 16 h with either 1 µg/ml tunicamycin or low amino acids (2.5% of DMEM). Results are representative of three biological replicates. Three technical replicates are shown. Unpaired, two-sided, nonparametric *t*-test: *$p < 0.05$, **$p < 0.005$, ***$p < 0.0005$. *p* values from top to bottom: 0.39, 0.0025, 0.015, 0.00049, 0.029. **e** Validation of DENR-dependent 5′UTRs predicted from ribosome footprinting. Graph shows 2–4 biological replicates for each reporter. ($n = 4$ for Lamin B1, Lamin B1 + stuORF, and c-Raf; $n = 3$ for a-Raf, PIK3R1, and PIK3R2; $n = 2$ for MAP2K6 and DROSHA) Unpaired, two-sided, nonparametric *t*-test: *$p < 0.05$, **$p < 0.005$, ***$p < 0.0005$, ****$p < 0.00005$. *p* values from top to bottom: 0.85, 0.000041, 0.0097, 0.0038, 0.00072, 0.0043, 0.31, 0.0081. **f** *DENR*[KO] and *eIF2D* knockdown synergistically decrease translation of the DENR-target reporters. Treatment with 1 µg/ml tunicamycin or low amino acids (2.5% of DMEM) for 16 h. Results are representative of two biological replicates. Three technical replicates are shown. Unpaired, two-sided, nonparametric *t*-test: *$p < 0.05$, **$p < 0.005$. *p* values from top to bottom: 0.0081, 0.00087, 0.032, 0.016, 0.98, 0.019, 0.00054, 0.00073, 0.54. **g** c-Raf and a-Raf protein levels are DENR dependent. Control or *DENR*[KO] cells were transfected with either GFP or DENR expression plasmid. Results are representative of three biological replicates. Source data, including uncropped western blots with molecular weight marker positions, are provided in the Source Data file.

The last two codons of a uORF could determine DENR•MCTS1 dependence because of the amino acids they code for. We assayed four synonymous alanine codons, and found that a uORF ending with GCG[Ala] is more DENR dependent than a uORF ending with GCC[Ala] (Supplementary Fig. 5c), suggesting the codon and not the amino acid is the relevant parameter. Consistent with this, mutation of the terminal GCG[Ala] codons in *ATF4* uORF2 or in *a-Raf* uORF1 to GCC[Ala] also blunted the DENR dependence (#4, Supplementary Fig. 5a, d). This is consistent with data from in vitro reconstituted translation systems[35] showing that the DENR•MCTS1 complex helps recycle post-termination 40S ribosomes, acting after the elongating peptide chain has been released, thereby making the sequence of the polypeptide irrelevant in this context.

From these findings we predict that mRNAs containing uORFs with GCG[Ala] or CTG[Leu] as the last two codons will be DENR dependent. To test this computationally, we first asked whether the uORFs upstream of DENR-target mORFs (Fig. 1c) are enriched for particular penultimate codons. We analyzed all ATG-initiated uORFs that are potentially translated in HeLa cells because they have at least one 80S ribosome footprint on their start codon ($n = 14892$). Of these, 1220 are in the 5′UTRs of DENR targets. In this group of 1220 uORFs, four penultimate codons were enriched, including ATG[Met] and GCG[Ala] (Supplementary Fig. 5e). Conversely, we asked whether mORFs of transcripts containing uORFs with individual penultimate codons are up- or downregulated genome-wide upon DENR loss-of-function. We grouped all mRNAs with potentially translated uORFs by uORF penultimate codon, and found that the penultimate codons CTG[Leu], ATG[Met], and GCG[Ala], but not GCC[Ala], were among the ones that correlated with reduced mORF TE upon DENR KO (Supplementary Fig. 5f). To test this experimentally, we selected mRNAs containing uORFs with penultimate GCG[Ala] or CTG[Leu] codons, that are also well translated based on publicly available Ribo-seq data, and cloned their 5′UTRs into luciferase reporter plasmids (Supplementary Fig. 5g). Indeed, two of the three tested 5′UTRs, of *CDK4* and *TSC1*, were DENR dependent (Fig. 3e). Consistent with this, CDK4 protein levels are DENR and eIF2D dependent (Fig. 3f, g).

**DENR•MCTS1 recycles certain post-termination 40S ribosomes.** DENR•MCTS1 complex has been shown to promote release of deacylated tRNA from post-termination 40S ribosomes, which are still bound to the mRNA and to one or two tRNAs in the P and E sites[35]. The identity of these two tRNAs is determined by the terminal codons of the ORF. Hence, we

hypothesized that DENR•MCTS1 are only required to process post-termination 40S ribosomes that contain certain tRNAs, such as the one decoding GCG[Ala] (i.e. in all other cases post-termination 40 ribosomes release tRNAs independently of DENR•MCTS1). Since eviction of the deacylated tRNA after a uORF is probably required for reinitiation of translation, this would explain why DENR•MCTS1 are required only after uORFs with certain terminal codons. To test this model, we analyzed post-termination 40S ribosome recycling genome-wide. If our model is correct, in the absence of DENR•MCTS1, post-termination 40S ribosomes should stall specifically on the stop codon of ORFs with penultimate codons that cause DENR dependence for reinitiation (e.g. GCG[Ala]). To this end, we employed 40S ribosome footprinting, which captures the positions of 40S ribosomes during scanning and recycling[47]. We and others recently adapted this method for animal cells[29,48,49]. This revealed that loss of DENR causes a strong accumulation of 40S recycling intermediates on stop codons of both main ORFs (mORFs) (Fig. 4a, length-resolved footprints Supplementary Fig. 6a) and uORFs (Fig. 4b, length-resolved footprints Supplementary Fig. 6b). Interestingly, the accumulation of 40S ribosomes on stop codons was highly variable between transcripts. For example, no 40S accumulation occurs on the stop codon of the *TUBA1B* mRNA (Fig. 4c), but a strong 40S accumulation occurs on the stop codon of the *RAN* mRNA (Fig. 4d). Quantification of 40S footprint density on the stop codon of main ORFs showed that 1222 of 6566 translated mORFs have a significant increase in *DENR*[KO] (by *z*-score analysis, >4-fold, Supplementary Data 2). In this set of mORFs, there is a strong enrichment for particular codons in the penultimate (−1) position, but not preceding codons (Supplementary Fig. 6c), indicating that the penultimate codon influences whether recycling 40S ribosomes stall upon DENR knockout. In agreement with our model, the penultimate codons that cause DENR-dependent reinitiation if they are present in uORFs—ATG[Met], CTG[Leu], and GCG[Ala]—all cause recycling defects when they are penultimate codons of mORFs (Fig. 4e). Likewise, out of 9982 uORF stop codons that had 40S ribosome footprints on them, we observed accumulation of 40S ribosomes on 891 of them (Supplementary Data 3). Among these, the penultimate codons CTG[Leu] and GCG[Ala] were again enriched (Fig. 4f and Supplementary Fig. 6d). In summary, our 40S footprinting data show that the penultimate codons of uORFs that cause DENR-dependent reinitiation are the same ones that cause defective 40S recycling both on main ORFs and uORFs upon DENR loss-of-function.

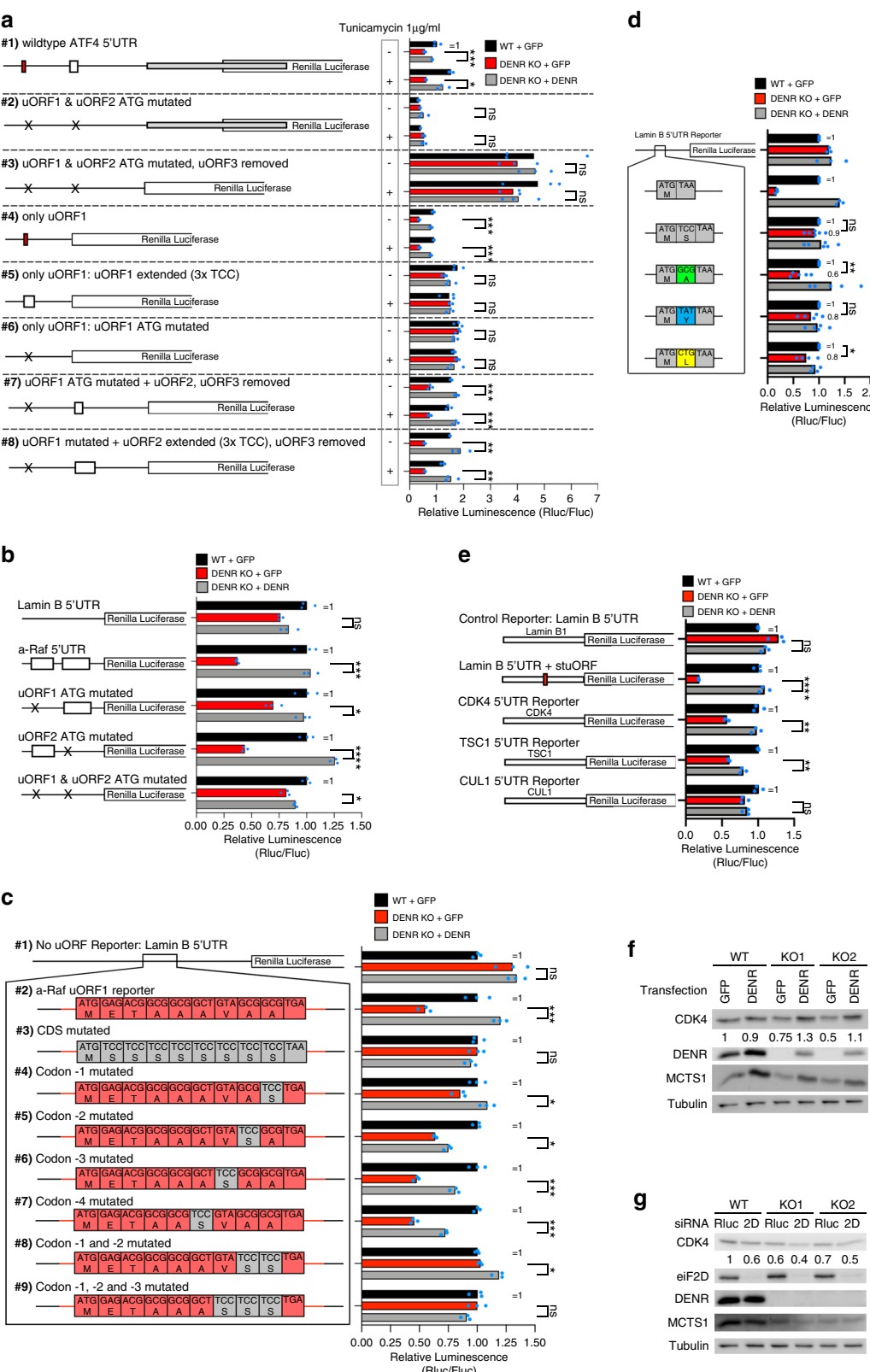

**DENR•MCTS1 expression correlates with ATF4 activity in cancer**. Since ATF4 helps cells cope with stress, it plays a central role in the survival of cancer cells, which are exposed to proteotoxic stress, hypoxia, nutrient stress, DNA damage, and oxidative stress. For instance, ATF4 is required for transformation of fibroblasts by H-rasV[12], it is essential for prostate cancer cells to grow and survive[23], and it has been implicated in chemotherapy resistance[50]. We therefore studied the functional consequences of DENR-dependent ATF4 translation. Reduced DENR and/or eIF2D expression lead to reduced proliferation of HeLa cells (Fig. 5a, b and Supplementary Figs. 2b, 7a). Since cells lacking DENR and eIF2D have very little ATF4 protein (Fig. 2c), we asked whether this contributes to their impaired proliferation. Indeed, ATF4 knockdown (Supplementary Fig. 7b) in HeLa cells

**Fig. 3 Penultimate codon identity determines DENR-dependent reinitiation after longer uORFs. a** uORF1 and uORF2 of the *ATF4* 5′UTR are DENR-dependent elements. Cells were treated with 1 µg/ml tunicamycin for 16 h. uORFs were mutated by changing the start codon to TAC. uORFs were extended by inserting three TCC$^{Ser}$ codons directly after the uORF start codon. Results are representative of three biological replicates. Three technical replicates are shown. Unpaired, two-sided, nonparametric *t*-test: *$p < 0.05$, **$p < 0.005$, ***$p < 0.0005$. *p* values from top to bottom: 0.00023, 0.0088, 0.31, 0.92, 0.28, 0.70, 0.00018, 0.00020, 0.15, 0.89, 0.16, 0.53, 0.00013, 0.00016, 0.0022, 0.0026. **b** The two long uORFs in the *a-Raf* 5′UTR impart DENR dependence. uORFs were mutated by changing the start codon sequence to TAC. Results are representative of three biological replicates. Three technical replicates are shown. Unpaired, two-sided nonparametric *t*-test: *$p < 0.05$, ***$p < 0.0005$, ****$p < 0.00005$. *p* values from top to bottom: 0.19, 0.00060, 0.0077, 0.0000053, 0.025. **c** The last two codons of *a-Raf* uORF1 determine DENR dependence. Results are representative of three biological replicates. Three technical replicates are shown. Unpaired, two-sided, nonparametric *t*-test: *$p < 0.05$, ***$p < 0.0005$. *p* values from top to bottom: 0.71, 0.00013, 0.30, 0.010, 0.0052, 0.00027, 0.00036, 0.0067, 0.13. **d** Introduction of GCG$^{Ala}$-Codon in a 2 amino acid uORF causes DENR-dependent reinitiation. Graph shows 3–6 biological replicates per reporter ($n = 6$ for TCC; $n = 5$ for GCG and TAT; $n = 4$ for CTG; $n = 3$ for Lamin B1 and Lamin B1 + stuORF). Unpaired, two-sided nonparametric *t*-test: *$p < 0.05$, **$p < 0.005$. *p* values from top to bottom: 0.30, 0.014, 0.30, 0.13. **e** DENR dependence of predicted DENR-target 5′UTRs as shown by translation luciferase reporters. Results are representative of three biological replicates. Three technical replicates are shown. Unpaired, two-sided nonparametric *t*-test: **$p < 0.005$, ****$p < 0.00005$. *p* values from top to bottom: 0.077, 0.00003, 0.00067, 0.0020, 0.48. **f** CDK4 protein levels are decreased by *DENR*$^{KO}$ and rescued by reconstitution with DENR. Control or *DENR*$^{KO}$ cells were transfected with either GFP or DENR expression plasmid, reseeded and harvested after 16 h. **g** DENR knockout and eIF2D knockdown decrease CDK4 protein levels. Control or *DENR*$^{KO}$ HeLa cells were transfected with siRNA pools targeting either GFP or eIF2D mRNA. Results are representative of two biological replicates. Source data, including uncropped western blots with molecular weight marker positions, are provided in the Source Data file.

dramatically impairs their proliferation (Fig. 5c), and over-expression of ATF4 in DENR KO cells (Supplementary Fig. 7c) partially rescues their proliferation (Fig. 5d). Nonetheless, reduced protein levels of additional DENR-target genes including Cdk4, a-Raf, and c-Raf, likely also contribute to the DENR phenotype. One important ATF4 target gene is *ASNS*[51]. Induction of ATF4 with tunicamycin leads to elevated *ASNS* transcription (Fig. 5e) and mildly elevated ASNS protein (lanes 1 and 7, Fig. 5f). (ASNS protein levels probably do not increase as strongly as the mRNA levels because tunicamycin also causes a global shutdown in translation). ASNS induction is reduced in *DENR*[KO] cells at the mRNA (Fig. 5e), and protein levels (lane 9, Fig. 5f). When eIF2D is co-depleted, *ASNS* mRNA and protein levels are further decreased (Supplementary Fig. 7d, e). Likewise, expression of ASNS during amino acid limitation is also DENR and eIF2D dependent (40–60% drop upon DENR/eIF2D double depletion, Supplementary Fig. 7f). Since leukemic cells express low ASNS, acute lymphoblastic leukemia (ALL) is treated with a combination of chemotherapy and asparaginase to deplete blood asparagine[52]. Patients develop ASNase resistance, however, when ALL cells upregulate ASNS expression[53], as seen in Jurkat cells (Fig. 5g). Knockdown of DENR or MCTS1 in Jurkat cells reduces ASNS protein levels (Fig. 5g), indicating DENR•MCTS1 are likely required for ASNase drug resistance in ALL.

More generally, ATF4 plays an important role in many different tumor entities[22–28,50]. Since we find DENR•MCTS1 is required for ATF4 expression in cervical cancer HeLa cells, leukemia Jurkat cells, and fibrosarcoma HT1080 cells, we postulated DENR•MCTS1 may promote ATF4 translation in many different cancers. To quantify this, we used *ASNS* mRNA levels as a readout for ATF4 activity. (Since ATF4 is mainly regulated translationally, *ATF4* mRNA levels are not a useful readout for ATF4 activity). Interestingly, *ASNS* mRNA levels correlate with DENR•MCTS1 levels in individual cancer entities such as liver hepatocellular carcinoma ($r > 0.35$), lung squamous cell carcinoma ($r > 0.5$) and renal cancer ($r > 0.49$, Supplementary Fig. 7g–i), and indeed throughout the entire TCGA pan-cancer set ($r > 0.36$, Fig. 5h). Likewise, a transcriptional signature of ATF4 activity comprised of 11 ATF4 targets[54] correlates with DENR•MCTS1 expression across all tumor entities ($r > 0.42$, Fig. 5i). In agreement with this, in cancer entities where ATF4 activity (assayed as *ASNS* mRNA levels) correlates with survival (e.g. renal cancer and liver hepatocellular carcinoma) (Supplementary Fig. 8a), also DENR or MCTS1 levels have prognostic value (Supplementary Fig. 8a).

Hence DENR•MCTS1 appear to promote ATF4 translation in a broad range of cancer cells.

## Discussion

We find here that DENR and MCTS1 are required in human cells to induce translation of ATF4 via translation reinitiation in response to stress. An accompanying paper from the Ryoo lab finds that these factors are also required for ATF4 translation in vivo in *Drosophila melanogaster*[55], indicating that this regulation is evolutionarily conserved from flies to humans. Hence DENR•MCTS1 play an important role in the ISR by enabling the central effector, ATF4, to be expressed. Through an unbiased ribosome footprinting screen, we identify a class of uORFs that require DENR•MCTS1 for efficient reinitiation. These uORFs are longer than one amino acid in coding sequence, but have certain specific penultimate codons including in particular GCG$^{Ala}$. Messenger RNAs containing such uORFs include those of *a-Raf, c-Raf, Cdk4,* and *PIK3R2*. In addition to *ATF4*, all of these target genes have been implicated in cancer and together may therefore explain why DENR and MCTS1 have oncogenic potential[40–43].

Why do uORFs ending with certain codons require DENR•MCTS1 for reinitiation? We find that the same codons cause a 40S recycling defect in the absence of DENR when they are present at the penultimate position of main ORFs. Our data do not allow us to distinguish between the role of the penultimate codon itself, and the role of the tRNA that decodes it. Nonetheless we propose a model of translation reinitiation whereby certain tRNAs, such as GCG$^{Ala}$, interact strongly with the 40S ribosomal subunit, and thereby require DENR•MCTS1 for their eviction after translation termination and 60S disjoining (Supplementary Fig. 8b). In this model, most other tRNAs disengage on their own, therefore not requiring DENR•MCTS1. Removal of the tRNA is necessary for the 40S to resume scanning, to re-recruiting an initiator tRNA via the ternary complex, and hence to reinitiate translation[56]. In agreement with this, different tRNAs dissociate from post-termination 40S ribosomes with different propensities[56]. The Pestova group has shown that some tRNAs, such as the C-TGC tRNA, spontaneously dissociate from terminating 40S ribosomes whereas others, such as the L-CTT tRNA, do not[56]. Interestingly, in our analysis of 40S footprints on main ORF stop codons, penultimate L-CTT is strongly enriched in the set of genes that have a 40S accumulation upon DENR KO, while penultimate C-TGC tRNA is strongly de-enriched. Hence the propensity of tRNAs to dissociate spontaneously from post-

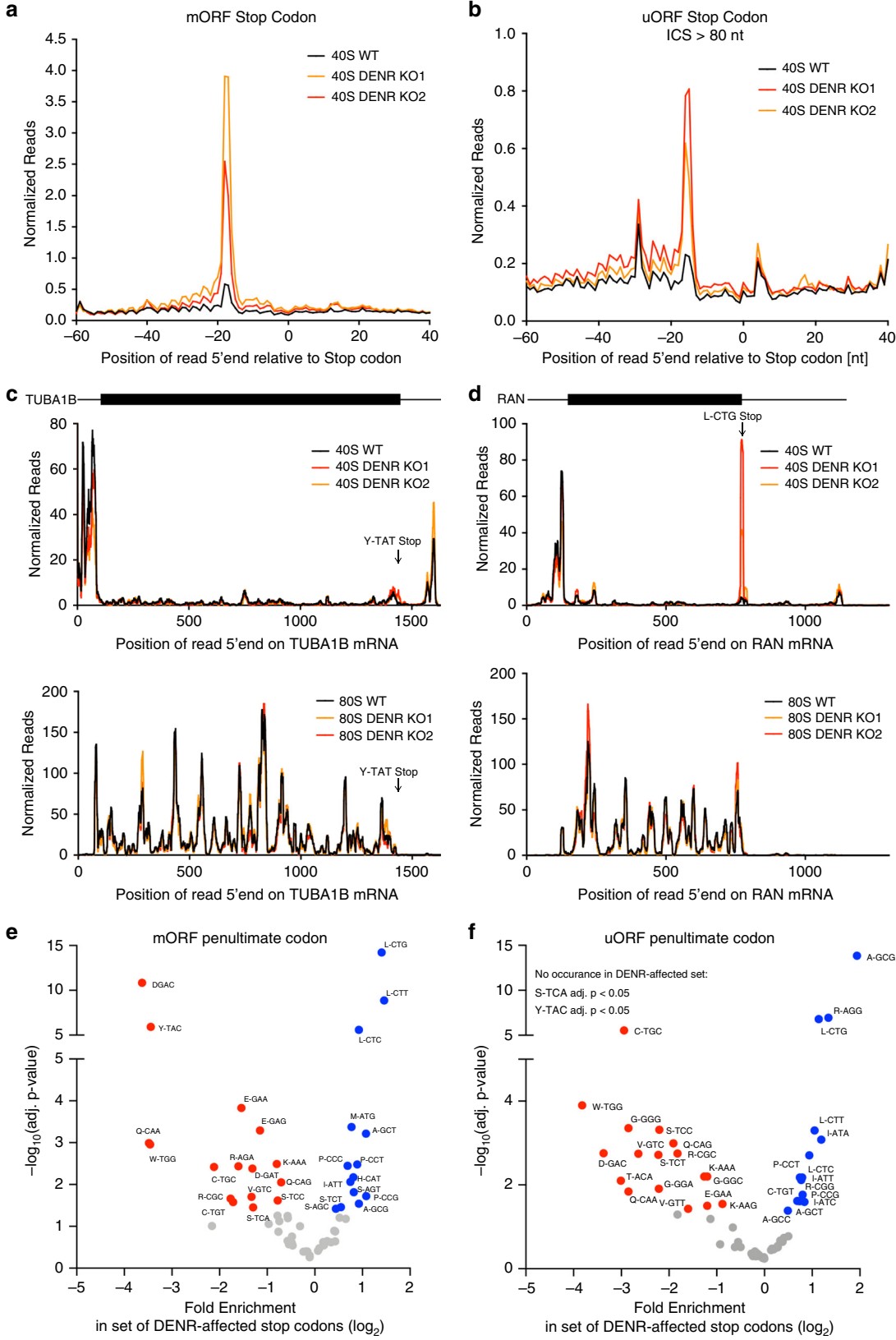

termination 40S ribosomes in vitro negatively correlates with the requirement of DENR in vivo. Also in agreement with this model, we find that most long uORFs, which have DENR-independent penultimate codons, support reinitiation as efficiently as DENR-dependent uORFs in the presence of DENR (Supplementary Fig. 1e). This suggests DENR is rescuing a reinitiation defect of a specific class of uORFs, and otherwise uORFs are generally permissive for reinitiation. If initiator tRNA interacts strongly with the 40S, then uORFs consisting of a start codon directly followed by a stop codon, which we described in the past as being DENR targets, would be a specific example of a uORF with a DENR-dependent penultimate codon.

**Fig. 4 DENR-dependent 40S ribosome recycling is affected by the penultimate codon transcriptome-wide. a, b** DENR facilitates 40S ribosome recycling after translation of **a** main ORFs (mORF) and **b** uORFs, as seen by an accumulation of 40S recycling intermediates on stop codons in the absence of DENR. Metagene profiles for 40S ribosome footprinting of control and DENR knockout HeLa cells showing the position of the 5′end of ribosome footprints relative to stop codons of **a** all protein coding ORFs or **b** all uORFs. Read counts were normalized to sequencing depth. **c** 40S ribosome recycling on the *TUBA1B* stop codon is not DENR-dependent. 40S (top panel) and 80S (bottom panel) ribosome occupancy on the *TUBA1B* transcript. Read counts were normalized to sequencing depth, and graphs were smoothened with 10 nt sliding window. 5′UTR features: 1 amino acid uORFs (red), longer uORFs (gray). **d** 40S ribosome recycling on the *RAN* mRNA is DENR-dependent. 40S (top panel) and 80S (bottom panel) ribosome occupancy on the *RAN* transcript. Read counts were normalized to sequencing depth, and graphs were smoothened with a sliding window of 10 nt. 5′UTR features: 1 amino acid uORFs (red), longer uORFs (gray). **e** Main ORFs that show an accumulation of 40S recycling intermediates in *DENR*^KO cells are enriched for specific codons in the penultimate position. Penultimate codon enrichments for mORFs that show 40S accumulation on the stop codon ($n = 1222$) relative to all detected mORF stop codons ($n = 6566$). Significance was assessed using binomial tests, adjusted for multiple testing. **f** uORFs that show an accumulation of 40S recycling intermediates in *DENR*^KO cells are enriched for specific codons in the penultimate position. Penultimate codon enrichments for uORFs that show 40S accumulation on the stop codon ($n = 891$) relative to all detected uORF stop codons ($n = 9982$) Statistical significance was assessed using binomial tests, adjusted for multiple testing. Source data are provided as a Source Data file.

Since main ORFs code for proteins with function, their penultimate codons may be dictated by functional requirements of the encoded protein. Most uORFs, however, are not thought to code for functional peptides, hence one could imagine that the penultimate codons would be free to mutate to one of the many codons that do not require DENR•MCTS1 for recycling. Hence it is tempting to speculate that the *ATF4* and *a-Raf* uORFs have DENR•MCTS1 dependent penultimate codons in many animal species (Supplementary Fig. 4b, c), for regulatory reasons.

The functional consequences of loss of DENR•MCTS1 in yeast and in animals are the opposite—loss of DENR in human or *Drosophila* cells leads to reduced expression of an ORF downstream of a uORF (Supplementary Fig. 1d), whereas in yeast it leads to increased expression[36]. Therefore, in yeast DENR•MCTS1 seem to only promote ribosome recycling and not translation reinitiation[36] whereas in human cells they do both. This may be because yeast ribosomes are generally non-permissive for reinitiation[57] whereas metazoan ribosomes are permissive[58–60]. In human cells, we observe that translation initiation factors such as eIF3, eIF4G1, and eIF4E are able to remain bound to elongating 80S ribosomes for circa 12 codons after the ATG[29]. As a consequence, after translating short uORFs ribosomes in human cells still have the necessary initiation factors for remaining associated with the mRNA and for a new round of initiation. This may be different in yeast. Yeast eIF3 consists of fewer proteins than human eIF3[61,62] and eIF4F binds the ribosome via eIF5 and eIF1[63–65], which leave the ribosome upon 60S subunit joining. Thus, it seems likely that in yeast eIF4F cannot persist on 80S ribosomes. Indeed, the best-characterized example of reinitiation in yeast occurs on the *GCN4* mRNA[66], which requires *cis*-acting features of the mRNA that stabilizes eIF3 on the terminating ribosome[67,68]. Hence, on uORFs eviction of the tRNA in post-termination 40S ribosomes by DENR may lead to the ribosome falling off the mRNA in yeast, whereas in human cells it may permit a renewed round of scanning and initiation. Likewise, on human main ORF stop codons where initiation factors are no longer present on the ribosome[29] DENR promotes recycling rather than reinitiation.

We find that DENR•MCTS1 levels correlate with *ASNS* mRNA levels, a readout for ATF4 activity, across the TCGA pan-cancer set. Due to the role of ATF4 in cancer, as well as the role of the other DENR-target genes identified here, this may explain why DENR•MCTS1 have been identified as oncogenes[40–42]. These results also impart the penultimate codons of uORFs in the transcriptome with regulatory potential, making this class of mRNAs dependent on DENR•MCTS1 and eIF2D for efficient translation.

## Methods

**Cloning.** Sequences of oligos used for cloning are provided in Supplementary Table 1. Plasmids for CRISPR/Cas9 mediated gene knockout were generated by annealing DNA oligos encoding the sgRNA and cloning them into the px459 vector (Addgene 48139) via the two Bbs1 sites. For DENR overexpression, pSS265 from[38] was used. For GFP and ATF4 overexpression, the respective coding sequences were cloned into the pcDNA 3.1 backbone via restriction sites EcoR1(GFP)/HindIII (ATF4) and Nhe1. The *Lamin B1* 5′UTR firefly and renilla luciferase reporters and the *Lamin B1* 5′UTR stuORF reporter were previously described[38]. The uORF length reporters (Supplementary Fig. 1e) were generated by oligo cloning the uORF sequences into the *Lamin B1* 5′UTR renilla luciferase reporter at the Spe1 and Age1 restriction sites. Renilla luciferase reporters with 5′UTRs of various genes (Figs. 2d–f and 3e) were cloned by amplifying the 5′ UTR of the gene of interest from HeLa cell cDNA and cloning it into the renilla luciferase reporter plasmid at the HindIII and Bsp119l sites. As the *ATF4* 5′UTR contains a BSP119l site, a Bsu15l to Bsp119l scar ligation was used for those reporters. Generally, uORFs were extended or deleted (Fig. 3a, b and Supplementary Fig. 4a) by site-directed mutagenesis PCR and cloned back into the original plasmid at the HindIII and BSP119l sites. The *ATF4* 5′UTR was shortened by PCR amplification of the fragment and insertion via Bsu15l and HindIII (Fig. 3a). Reporters with uORFs in the *Lamin B* 5′UTR context (Fig. 3c, d and Supplementary Fig. 5a–d) were generated by oligo cloning the uORF sequences into the *Lamin B1* 5′UTR renilla luciferase reporter at the Spe1 and Age1 restriction sites.

The *ATF4* 5′UTR reporter used in this paper contains the first 364 nucleotides of the ATF4 mRNA (NM_182810.2), which include the 5′UTR (281 nt) and the ATF4 ORF up to the end of uORF3, fused to the Renilla luciferase coding sequence so that the ATF4 mORF is in frame with the Renilla luciferase. Since the transcription start site for all reporters including the normalization control is within the CMV promoter region, all reporters contain an additional 18 nt leader sequence at the 5′ end up to the first available restriction site (gtcagatcactagaagct).

**Cell culture.** HT1080 cells were kindly provided by Stefan Pusch. HeLa (ATCC) cells and HT1080 (ATCC) cells were cultured in DMEM +10% fetal bovine serum +100 U/ml Penicillin/Streptomycin (Gibco 15140122). Cells were subcultured using Trypsin-EDTA for dissociation. Cellular stress conditions were induced by treatment with tunicamycin at 1 μg/ml, or by amino acid starvation at 2.5% of the normal DMEM amino acid concentration (using dialyzed FBS and amino acid free DMEM), or with sodium arsenite (250 nM). Jurkat (ATCC) cells were cultured in RPMI, 10% FBS, 100 U/ml Penicillin/Streptomycin (Gibco 15140122).

For plasmid expression, cells were transfected with Lipofectamine 2000. Cells were seeded at 700.000 (HeLa controls) or 900.000 (HeLa *DENR*^KO) cells per 10 cm dish and transfected with 6 μg plasmid DNA + 12 μl Lipofectamine reagent. After 5 h, medium was exchanged and after 24 h cells were reseeded for experiments. For the ATF4 overexpression experiment (Fig. 5d and Supplementary Fig. 7c), cells were transfected directly in the 96-well format at low cell concentration, since reseeding caused a loss of ATF4 overexpression. For this, Lipofectamine 3000 was used as it causes less cell toxicity: control and DENR KO cells were seeded at 1000 and 1200 cells per well, respectively. On the next day, cells were transfected with 50 ng DNA, 0.1 μl P3000, and 0.075 μl Lipofectamine 3000 per well. Medium was exchanged after 4 h. The proliferation curve was performed by harvesting Plate 0 on the same day, and one plate every successive day, assayed with CellTiter-Glo.

For siRNA mediated knockdowns, cells were transfected with Lipofectamine RNAimax. Cells were seeded at 200.000 cells per six-well, transfected with 3 μl 10 μM siRNA and 9 μl Lipofectamine reagent. Cells were reseeded for experiments 48–72 h after transfection. Sequences of siRNAs are provided in Supplementary Table 2.

*DENR* and *MCTS1* KO cells were generated by CRISPR/Cas9 editing. Cells were transfected with the pX459 plasmid coding for Cas9, for sgRNAs targeting *DENR* or *MCTS1*, and for puromycin resistance. After 24–48 h, transfected cells were

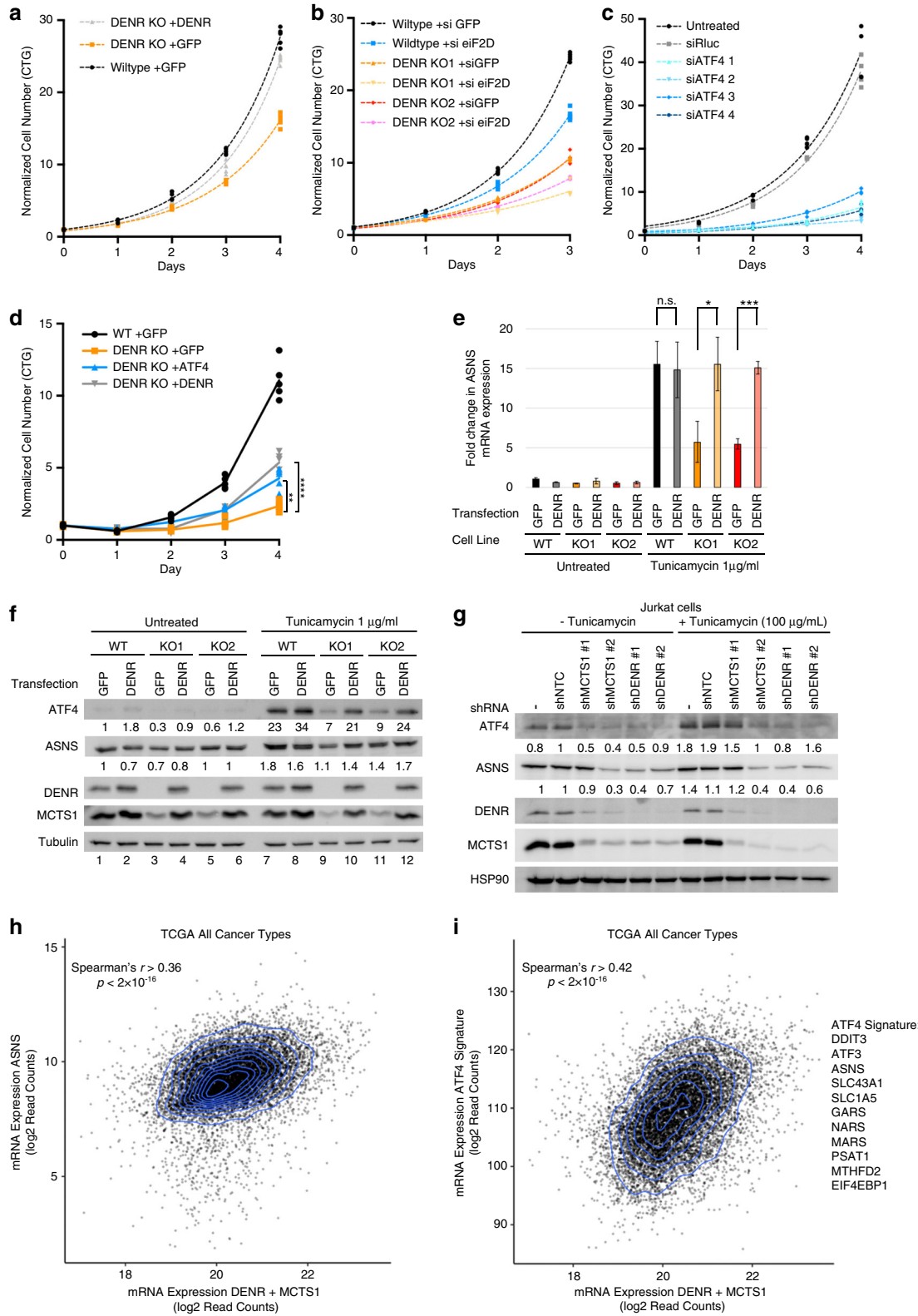

selected using 1 µg/ml puromycin for 2–3 days. Then cells were grown out and seeded at limiting dilution and clones were screened for successful knockout by immunoblotting. Successful knockout was then also assessed by sequencing the edited genomic region after PCR amplification and TOPO cloning.

**Lentiviral transduction.** Lentiviral plasmids were generated by introducing the following sequences into the Cellecta backbone pRSI21-U6-sh-HTS6-CMV-TagGFP2-2A-Puro by oligo cloning: nontarget shRNA (CAACAAGATGAAG

AGCACCAA), two shRNA constructs targeting DENR (shDENR #1: GATGA TGACAGCATCGAAGAT; shDENR #2: GAATTAGTGAGGGTCAAGGAA) and two shRNA constructs targeting MCTS1 (shMCTS1 #1: CCCTAAGATTAC TTCACAAAT; shMCTS1# 2: TGTACTCAGTGGAGCAAATAT). 293T cells were seeded in 10-cm dishes and co-transfected with these plasmids, together with pMD2.G and psPAX2 using Trans IT transfection reagent (MirusBio, MIR 2300). Lentiviruses were harvested 72 h post-transfection and filtered through a 0.45 µm PVDF sterile syringe filter (Carl Roth GmbH, P667.1). For Jurkat cells, 50,000 cells were supplemented with 5 µg/ul of polybrene (Merck Millipore, RT-1003-G) and

**Fig. 5 DENR/MCTS1 and eIF2D promote ATF4 activity in cancer. a** DENR is required for optimal proliferation of HeLa cells. Control and *DENR*^KO HeLa cells were transfected with GFP or DENR expression plasmids. Proliferation was assayed using CellTiter-Glo. Results are representative of three biological replicates. **b** DENR and eIF2D inhibit proliferation in an additive manner. Control and *DENR*^KO HeLa cells were transfected with siRNAs targeting GFP or eIF2D, proliferation was assayed using CellTiter-Glo. Results are representative of three biological replicates. **c** ATF4 is required for optimal proliferation of HeLa cells. ATF4 protein was depleted using four independent siRNAs targeting *ATF4*. Cell proliferation was assayed using CellTiter-Glo. Results are representative of three biological replicates. **d** DENR promotes proliferation in part through ATF4. Control and *DENR*^KO HeLa cells were transfected with GFP, DENR, or ATF4 expression plasmids. Proliferation was assayed using CellTiter-Glo. Results are representative of three biological replicates. **e, f** DENR is required for efficient induction of the ATF4 target gene *ASNS*. **e** Q-RT-PCR of *ASNS* and *ATF4* mRNA levels normalized to *Actin B* mRNA ($n = 3$ technical replicates). **f** Western blot analysis of control or *DENR*^KO HeLa cells treated with 1 μg/ml tunicamycin for 16 h. Cells were additionally transfected with GFP or DENR expression plasmids as indicated. Quantification of ATF4 and ASNS band intensity is normalized to tubulin. Cells from the same experiment as in **e**. Data are presented as mean values ± SD of three technical replicates. Unpaired, two-sided, nonparametric *t*-test: *$p < 0.05$, ***$p < 0.0005$. *p* values from left to right: 0.79, 0.016, 0.000096. Results are representative of three biological replicates. **g** ATF4 and ASNS expression is DENR dependent in Jurkat cells. Cells were transduced with virus encoding shRNAs targeting *DENR* or *MCTS1* and after 7 days harvested for western blotting, tunicamycin treatment at 1 μg/ml for 16 h. Results are representative of three biological replicates. **h** *ASNS* mRNA levels, as a readout for ATF4 activity, correlate to *DENR* + *MCTS1* mRNA levels across all cancers. Correlation, using Xena[73], of TCGA mRNA expression data. Shown is spearman's *r* and *p* value. $n = 11060$. *p* value $<2.2 \times 10^{-16}$. **i** An 11-gene transcriptional signature for ATF4 activity[54] correlates to *DENR* + *MCTS1* mRNA levels across all cancers. Correlation, using Xena[73], of TCGA mRNA expression data. Shown is spearman's *r* and *p* value. $n = 11060$. *p* value $<2.2 \times 10^{-16}$. Source data, including uncropped western blots with molecular weight marker positions, are provided in the Source Data file.

---

reverse transduced by spinoculation for 45 min at 800 g and at 32 °C. 7 days post-transduction Jurkat cells were centrifuged for 3 min at 200 × *g*, washed, and harvested for western blotting.

**Immunoblotting**. Cells were lysed using standard RIPA lysis buffer containing protease inhibitors (Roche mini EDTA-free, 1 tablet in 10 ml) and phosphatase inhibitors (2 mM Sodium Ortho-Vanadate, Roche Phosstop 1 tablet in 10 ml, 0.1 M Sodium Fluoride, 0.1 M beta-Glycerophosphate) and Benzonase (50 U/ml), after washing briefly with FBS-free DMEM. Lysates were clarified and protein concentration was determined using a BCA assay. Equal protein amounts were run on SDS-PAGE gels and transferred to nitrocellulose membrane with 0.2-μm pore size. After Ponceau staining membranes were incubated in 5% skim milk PBST for 1 h, briefly rinsed with PBST and then incubated in primary antibody solution (5% BSA PBST or 5% skim milk PBST) overnight at 4 °C. Membranes were then washed three times, 15 min each in PBST, incubated in secondary antibody solution (1:10,000 in 5% skim milk PBST) for 1 h at room temperature, then washed again three times for 15 min. Finally, chemiluminescence was detected using ECL reagents and the Biorad chemidoc. No membranes were stripped. Antibodies used in this study are listed in Supplementary Table 3.

For samples treated with Lambda phosphatase, lysis was carried out in Lysis buffer (0,25 M HEPES pH 7.5, 50 mM MgCl₂, 1 M KCl, 5% NP40), without phosphatase inhibitors. After lysates were clarified, 10 μl 10× NEBuffer for Protein Metallo Phosphatases (PMP) was added to 90 μl Lysate and 2 μl NEB Lambda Protein Phosphatase (Lambda PP) was added. Solutions were incubated at 30 °C for 30 min. Then, protein concentration was determined by BCA assay and sample was diluted in 5× Laemmli buffer and boiled.

**Quantitative RT-PCR**. Total RNA was isolated from cells using TRIzol following manufacturer's protocol. To synthesize cDNA, 1 μg of total RNA was used for oligo dT primed reverse-transcription using Maxima H Minus Reverse Transcriptase. Quantitative RT-PCR was done with Maxima SYBR Green/ROX mix on a StepOnePlus Real-Time PCR System. Actin B was used as a normalization control and all samples were run in technical triplicates. Sequences of Q-RT-PCR oligos used in this study are in Supplementary Table 4.

**Translation Reporter Dual-luciferase Assay**. Translation reporter assays after siRNA mediated knockdowns were carried out as follows: Cells were transfected with siRNAs as described above. After 72 h of transfection, cells were reseeded into a 96-well plate. HeLa control cells were seeded at 8.000 cells per well, *DENR*^KO cells were seeded at 12.000 cells per well. Cell were transfected 16–20 h after reseeding using Lipofectamine 2000. In total, 100 ng of renilla luciferase plasmid and 100 ng of firefly luciferase plasmid was used per well. After 4 h of transfection, the medium was exchanged.

For translation reporter assays including a DENR reconstitution condition, HeLa control cells were seeded at 8.000 cells per 96-well, while *DENR*^KO HeLa cells were seeded at 12.000 cells per 96-well. After 16–20 h of seeding, these cells were transfected with three plasmids using lipofectamine 2000. Per well, 60 ng of either GFP or DENR expression plasmid, 70 ng of renilla luciferase reporter plasmid, and 70 ng of firefly reporter plasmid were used. After 4 h of transfection, the medium was exchanged.

Renilla luciferase plasmids always contained the 5′UTR of interest, while the firefly luciferase plasmid always contained the DENR-independent *Lamin B1* 5′ UTR and was used as a transfection normalization control. A total of 0.4 μl Lipofectamine reagent was used per 96-well. Cells were always transfected in

triplicates. After 16–20 h of transfection, luciferase activity was assayed using the Dual-Luciferase® Reporter Assay System by Promega according to the manufacturer's instructions.

**Ribosome footprinting**. HeLa control and *DENR*^KO cells were seeded in 15 cm dishes at 3 million cells per dish in 20 ml growth medium. Two days later, cells were harvested for Ribo-seq and RNA-seq. For RNA-seq, cells were lysed with TRIzol and total RNA was isolated following manufacturer's protocol. For Ribo-seq, cells were briefly rinsed with ice-cold PBS containing 10 mM MgCl₂, 400 μM cycloheximide (CHX). This solution was poured off, removed by gently tapping the dish onto paper towels, and cells were lysed with 150 μl of lysis buffer (0.25 M HEPES pH 7.5, 50 mM MgCl₂, 1 M KCl, 5% NP40, 1000 μM CHX) per plate. Cells were scraped off and lysate was collected. After brief vortexing, lysate was clarified by centrifuging for 10 min at 20,000 × g at 4 °C. Approximate RNA concentration was measured using a Nanodrop system and 100 U of Ambion RNAse 1 was added per 120 μg of measured RNA. To prepare undigested polysome profiles (Supplementary Fig. 3b), RNAse was simply omitted. Lysates were incubated with RNAse for 5 min on ice. Lysates were then pipetted onto 17.5–50% sucrose gradients, which were produced by freezing and layering 50% (2.5 ml), 41.9% (2.5 ml), 33.8% (2.5 ml), 25.6% (2.5 ml), and 17.5% (1.8 ml) sucrose solutions (10 mM Tris HCl pH 7.4, 10 mM MgCl₂, 140 mM KCl, 200 μM CHX) in Seton Scientific Polyclear Tubes 9/16 × 3-3/4 IN, and centrifuged at 35,000 rpm for 3.5 h in Beckmann SW40 rotor. Gradients were fractionated using a Biocomp Gradient Profiler system and 80S fractions were collected for footprint isolation. RNA was isolated from these fractions using Acid-Phenol extraction and analyzed on an Agilent Bioanalyzer system to asses RNA integrity.

**40S ribosome footprinting**. This method was performed as described in ref. [29]. Two days before cell harvest, HeLa cells were seeded at 3 million cells per 15 cm dish in 20 ml growth medium. After brief washing, freshly prepared crosslinking solution (1× PBS, 10 mM MgCl₂, 400 μM Cycloheximide, 0.025% PFA, 0.5 mM DSP) was added to the cells. Cells were incubated with crosslinking solution for 15 min at room temperature while slowly rocking. Crosslinking solution was then poured off and remaining crosslinker was inactivated for 5 min with ice-cold quenching solution (1× PBS, 10 mM MgCl₂, 200 μM Cycloheximide, 300 mM Glycine). Quenching solution was poured off and 150 μl of lysis buffer (0.25 M HEPES pH 7.5, 50 mM MgCl₂, 1 M KCl, 5% NP40, 1000 μM CHX) was added to each 15 cm dish. Lysis was carried out at 4 °C. Cells were scraped off the dish and lysate was collected. After brief vortexing, lysates were clarified by centrifugation at 20,000 × g for 10 min at 4 °C. Supernatant was collected and approximate RNA concentration was determined using a Nanodrop photo-spectrometer. Overall, 100 U of Ambion RNAse 1 was added per 120 μg of measured RNA. Lysates were incubated for 5 min at 4 °C and then loaded onto 17.5–50% sucrose gradients and centrifuged for 5 h at 35,000 rpm in a Beckman Ultracentrifuge in the SW40 rotor. Gradients were fractionated using a Biocomp Gradient Profiler system. 40S and 80S fractions were collected for immunoprecipitation and footprint isolation. 40S and 80S fractions corresponding to roughly one or two 15 cm dishes were used for direct extraction of RNA for total footprint samples. Footprint fractions were then subjected to crosslink removal and RNA extraction: 55 μl (1/9th of volume) of crosslink-removal solution (10% SDS, 100 mM EDTA, 50 mM DTT) was added, 600 μl Acid-Phenol Chloroform (Ambion) was added and mixture was incubated at 65 °C, 1300 rpm shaking for 45 min. Tubes were then placed on ice for 5 min, spun for 5 min at 20,000 × g and supernatant was washed once with acid-phenol chloroform and twice with chloroform, then RNA was precipitated with

Isopropanol and subjected to library preparation (see below). The organic phase was used to isolate the precipitated or total proteins. In total, 300 μl ethanol were added, then 1.5 ml isopropanol were added and solutions were incubated at −20 °C for 1 h. Proteins were sedimented by centrifugation at 20,000 × g for 20 min, washed twice with 95% ethanol 0.3 M guanidine HCl, dried and resuspended in 1× Laemmli buffer.

**Deep-sequencing library preparation**. For 80S footprinting and RNA-seq libraries were prepared as follows:

RNA samples were depleted of ribosomal RNA using the Illumina Ribo-Zero Gold kit. Depleted total RNA was then fragmented using chemical cleavage in 50 mM NaHCO₃ at pH 10, 95 °C for 12 min. Then total RNA was processed in parallel with the depleted RNA from 80S ribosome fractions. For size selection, RNA was run on 15% Urea-Polyacrylamide gels and fragments from 25 to 35 nt were excised using reference ssRNA nucleotides of 25 and 35 bp run on a neighboring lane. RNA was extracted from the gel pieces and phosphorylated using T4 PNK. Deep-sequencing libraries were prepared from these RNA fragments using the Bio-Scientific NEXTflex Small RNA-Seq Kit v3. DNA was amplified with nine PCR cycles for the Ribo-seq samples and 11–13 cycles for the RNA-seq samples. Deep-sequencing libraries were sequenced on the Illumina Next-Seq 550 system.

40S footprinting libraries were prepared as follows:

After RNA extraction from 40S fractions, RNA quality and integrity were determined on an Agilent Bioanalyzer using the total RNA Nano 6000 Chip. For size selection, RNA was run on 15% Urea-Polyacrylamide gels (Invitrogen) and fragments of size 20–80 nt were excised using the Agilent small RNA ladder as a reference. RNA was extracted from the gel pieces by smashing the gels into small pieces with gel smasher tubes and extracting the RNA in 0.5 ml of 10 mM Tris pH 7 at 70 °C for 10 min. Gel pieces were removed and RNA was precipitated using Isopropanol. Footprints were then dephosphorylated using T4 PNK (NEB) for 2 h at 37 °C in PNK buffer without ATP. Footprints were then again precipitated and purified using isopropanol. Footprints were then assayed using an Agilent Bioanalyzer small RNA chip and Qubit smRNA kit. Overall, 25 ng or less of footprint RNA was used as input for library preparation with SMARTer smRNA-Seq Kit for Illumina from Takara/Clontech Laboratories according to the manufacturer's instructions. Deep-sequencing libraries were sequenced on the Illumina Next-Seq 550 system.

**Data analysis and statistics**. Analysis of ribosome footprinting NGS data: adapter sequences and randomized nucleotides were trimmed from raw reads using cutadapt (https://doi.org/10.14806/ej.17.1.200)[69]. Ribosomal RNA and tRNA reads were removed by alignment to human tRNA and rRNA sequences using bowtie2[70]. Then, the remaining reads were separately aligned to the human transcriptome (Ensembl transcript assembly 94) and human genome (hg38) using BBmap (https://sourceforge.net/projects/bbmap/)[71]. Generally, counted reads were normalized to sequencing depth (number of alignments per library). Read counting, metagene plots (Figs. 1a, b, 4a, b, and Supplementary Fig. 6a, b) and single transcript traces (Fig. 1d, f and Supplementary Figs. 2e, h, i, 4c, d) were carried out or created with custom software written in C available on GitHub (https://github.com/aurelioteleman/Teleman-Lab). TE was calculated from the number of 80S ribosome footprints in a coding sequence divided by the number of RNA sequencing reads on the transcript.

uORFs were annotated by the presence of an ATG and an in-frame stop codon inside the 5′UTR of any mRNA. Only ATG-initiated uORFs were considered.

For Fig. 1c, changes in TE were calculated as $log_2 \left( \frac{TE\,DENR\,KO}{TE\,WT} \right)$. Theoretical Z vs. Observed Z analysis was carried out to determine transcripts with changes in TE higher than expected from gaussian distribution. For the final presentation of the data in Fig. 1c, only the most strongly affected transcript variant of each gene was considered. GO enrichment analysis (Supplementary Fig. 2d) was carried out using the list of downregulated mRNAs from the Z vs. Z analysis and a list of all detected mRNAs as a background list using the Panther GO enrichment analysis platform[72].

For Fig. 1e, groups containing uORFs of certain lengths are nonexclusive, therefore each transcript can be part of multiple groups. Each individual transcript variant for each gene was assigned to groups. Transcripts containing 1AA uORFs were excluded from all other uORF groups to ensure that reduction in TE in these groups was not due to 1AA uORFs. Multiple mapping of ribosome footprints was allowed to account for transcript variants. Statistically significant differences between mRNA groups were assessed by nonparametric Kruskal–Wallis test corrected for multiple comparisons.

For Supplementary Figure 2c, human orthologs of mouse genes were determined by conversion of mouse gene ID to the human orthologue via the NCBI Homologene database.

DENR dependence of ribosome recycling on individual stop codons from 40 S footprinting data was calculated as follows: The number of footprints in a 10 nt window at the stop codon peak was counted, normalized to sequencing depth and the log2(FC) in reads in this area between DENR KO and WT cells was calculated. The log2(FC) was averaged between the two DENR KO cell lines and then a Z-Score analysis was used to classify stop codons into DENR-dependent versus independent. The frequency of terminal codons in DENR-dependent stop codons was compared to the frequency in all detected stop codons to determine codon

enrichments ($log_2 \left( \frac{Frequency\,DENR-dependent}{Frequency\,All\,detected} \right)$) (Fig. 4e, f and Supplementary Fig. 6c, d). Statistical significance of enrichments was assessed using binomial testing. $P$ values were adjusted for multiple testing (61 tests, one for each amino acid coding codon).

uORFs were classified as potentially translated when 80S ribosome footprints were detected in a 10 nt window on the putative start codon peak. Penultimate codon enrichments in translated uORFs present in the list of 517 DENR-targets (Supplementary Fig. 5e) were then calculated and statistically assessed as described above. The potential effect on translation efficiency by each uORF penultimate codon (Supplementary Fig. 5f) was calculated by assigning the log2FC(TE) of each transcript to all potentially translated uORFs on that transcript.

For luciferase assays, statistically significant differences were determined using nonparametric, unpaired, two-sided $t$-tests.

For immunoblots, all band quantifications shown in the figures are normalized to loading controls.

DENR, MCTS1 mRNA expression, and ASNS mRNA expression of ATF4 target signature mRNA expression data from TCGA was downloaded from the UCSC Xena browser (https://doi.org/10.1101/326470)[73]. Spearman correlation coefficients and significance were calculated in Graphpad Prism.

**Reporting summary**. Further information on research design is available in the Nature Research Reporting Summary linked to this article.

## Data availability

The datasets generated during the current study are available at NCBI Geo (accession GSE140084). Source data are provided with this paper. All other datasets generated and/or analyzed during the current study are available from the corresponding author on reasonable request.

## Code availability

All custom software used in this study is available at GitHub: https://github.com/aurelioteleman/Teleman-Lab.

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

## Acknowledgements

We thank Georg Stoecklin and Johanna Schott for sharing with us their ribosome footprinting protocol and Francesca Tuorto for discussion and input regarding tRNAs. This work was funded in part by a Deutsche Forschungsgemeinschaft (DFG, German Research Foundation) grant (project-ID 201348542, SFB 1036) to B.B and A.A.T., a DFG grant (316695455) to A.A.T., a DKFZ NCT3.0 Integrative Project in Cancer Research

(NCT3.0_2015.54 DysregPT) grant to B.B. and A.A.T., and Cell Networks—Cluster of Excellence (EXC81) grants to J.B. and K.C.v.H. High-throughput sequencing was carried out at the DKFZ Genomics and Proteomics Core Facility, or using an Illumina Next-Seq 550 system funded by the Klaus Tschira Foundation gGmbH, Heidelberg, Germany. The results shown in Fig. 5h, i and Supplementary Figs. 7g–i, 8a are based upon data generated by the TCGA Research Network: https://www.cancer.gov/tcga.

## Author contributions

Experiments were performed by S.B. (Fig. 5g); L.H. (Figs. 2d, 3a, 5c and Supplementary Figs. 3c, 7b) and J.B. (all others). J.B., L.H., S.B., K.C.vH., K.F., G.K., B.B., and A.A.T designed the work, analyzed data, interpreted data, and wrote the paper.

## Funding

## Competing Interests

The authors declare no competing interests.
