## [Peer Review File · Nature Communications]

Reviewers' comments:

Reviewer #1 (Remarks to the Author):

The manuscript by Bohlen et al. analyzes the regulation of ATF4 and other oncogenes through uORFs that operate through reinitiation involving DENR-MCTS1. First, the authors identified endogenous DENR-regulated transcripts in HeLa cells by ribosome profiling (Figure 1) and observed that these carried not only short, but also long uORFs. Of note, the same authors' previously published studies (caveat: these were not based on endogenous targets but mainly on reporters) had postulated that DENR operated mainly via short and even ultrashort (1 aa) uORFs. Among the most affected endogenous targets were ATF4 (well-known for its stress-dependent uORF regulation) and some other oncogenes. In Figure 2, the authors systematically and diligently dissect the requirements of ATF4, c-Ras and a-Raf translation for uORF regulation by DENR-MCTS1; in addition, they uncover a role for eIF2D in reinitiation events. In Figure 3, the dissection goes further in detail leading to the interesting finding that for the examples tested, the ultimate/penultimate codon identity on long uORFs determines DENR-dependent reinitiation. Finally (Figure 4), the authors try to link the observed regulation to cancer. The authors present experiments on cell growth in the presence/absence of DENR, eIF2D and ATF4, and then focus mainly on one ATF4 target, asparagine synthetase (ASNS). ASNS levels show correlation with DENR expression across cancers.

This is an interesting study that contains a lot of data and that is well designed and presented, in particular for the first three-quarters (Figures 1-3). The most interesting point is the ability of certain codons at -2 and -1 positions within uORFs to render them DENR-dependent. An important task for the revision should lie in further analyzing this finding by metatranscript analyses around the uORF regions in order to determine if such codons correlate with DENR-dependent translational activity genome-wide (i.e. beyond the examples).

I am a bit less impressed by the relevance of Figure 4 -- the link to ATF4 expression and cancer is somewhat weak. I believe DENR and alike were already shown to have prognostic value in cancer in recent publications. The link with ATF4 provides some novelty, but in order to go further than simple correlations, it would be important to show, as a minimum, rescue of the growth/survival phenotype of DENR knockdown by ATF4 expression. A few other points are listed below as well.

Main points:

1) One of the novel, interesting findings of the manuscript is the importance of certain codons at penultimate and final positions of uORFs in determining DENR dependency. While the extensive testing of different codon combinations using the dual-luciferase system elegantly demonstrates the specificity of GCG/CTG codons for the hand-picked examples, it is not clear to what extent these codons are used and can be associated with DENR-dependent regulation transcriptome-wide. Were these particular codons (or maybe others) enriched at the penultimate and ultimate positions within the transcript group regulated by DENR (e.g. the 518 down-translated genes from Fig. 1, or a similar set of genes). A metagene analysis on the landscape of all uORFs should show an enrichment of such codons correlating

with down-regulation in DENR-KO.

2) The proliferation defect in DENR-KO cells (Figure 4A-B) and siATF4 samples is quite striking, since ATF4 is anyway not highly expressed in cells. Given that DENR and EIF2D have hundreds of direct targets, it is not clear if the loss-of-function effect really comes from ATF4, as postulated. It is therefore important to show that the phenotype in DENR-KO/si eIF2D can be rescued by expressing additional ATF4. This is particularly important as the authors claim that the DENR phenotype could be due to low ATF4 expression. However, a direct positive test is missing.

3) First point regarding the 1-aa uORFs: There is in general a bit of overexcitement about the finding that not only 1-aa long uORFs but longer ones are involved in DENR dependent regulation as well, as if this is an unexpected and very new finding. As far as this reviewer understands, the 1-aa finding stemmed largely from reporter assays from the lab's previous studies. Other recent papers did not confirm the "short uORF" hypothesis (e.g. PMID 30982898). All this is not presented or discussed in a sufficiently balanced fashion. The novelty of the finding should thus be toned down, and the interpretation and reporting should be adapted accordingly.

4) Second point regarding 1-aa uORFs: In Fig 1C, the authors describe that Droscha displayed increased ribosome density on the 1-aa uORFs. However, this is very difficult to observe in this Figure due to several reasons: 5-nt windowing creating a diffused signal distribution, insufficient zoom on the X-axis and close proximity of 1-aa uORFs and longer ones to each other. Furthermore, in Fig 1A, the 5'-end of the reads were used for mapping (not A-, or P-site corrected positions) which shifts the signal approximately 15 nt upstream. However, in 1C, some of the strongest peaks can be observed downstream of the 1-aa uORF sites. To properly distinguish if the ribosome footprint densities are overlapping specifically with the 1-aa uORFs a more precise computational approach at least at codon resolution is necessary.

5) Third point regarding 1-aa uORFs: In Fig 1D, it was reported that transcripts containing one or more 1-aa uORF were excluded from all other uORF groups. However, it is not clear if the 1-aa uORF group contained also longer uORFs. Also, size (number of transcripts) in each group should be reported. How were genes with multiple transcript models handled? Was same uORF used in different groups? A negative correlation between uORF size and number of non-overlapping uORFs is expected; was this accounted for?

6) This manuscript adds to a number of studies that identify DENR targets – e.g. Schleich et al. 2014 and 2017 (the authors' lab) and Castelo-Szekely et al. 2019 (PMID 30982898). The latter also suggested ATF4/5 as targets and identified several cancer-related genes (Klhdc8a, Map2k5). Can the authors discuss determine and discuss the overlap between the studies (e.g. was Atf4 also a target in Drosophila; is Droscha a target across species/systems?).

7) Fig. 1B it is not immediately clear from which values the Z-scores were calculated – change in log-

transformed TE or simply change in TE? Was a test of significance performed which is well adapted for TEs (e.g. Xtail)? How many of these 518 were significantly down-translated in DENR-KO?

8) In general, annotation of uORFs is not sufficiently communicated. Were ribosome profiling data utilized to identify translated uORFs? Were non-canonical start codons allowed, or just ATG, in identification of putative uORF positions?

9) Fig S4A #9 does not agree with Fig 3C #4. In Fig 3C, Codon-1 GCG>TCC mutation seems to have lost DENR-dependence while in Fig S4A, the same mutation is clearly DENR dependent. Does it imply variability within the assay, or certain codons? Can the authors add more biological replicates and discussion to clarify such cases?

10) Some Western Blots lack quantification (for example Fig 4G). Are the quantifications already normalized to loading control?

Other comments:

11) Intro + abstract: The 1000-fold change in TE has not been reported in any empirical study. The Schwannhausser paper that is cited infers translation from a model. The authors may want to use a different reference that actually shows factor 1000, or put this value down to more common estimates that are rather 10-fold (e.g. ES-cell data, Ingolia et al. 2011, PMID 22056041) to 100-fold (e.g. budding yeast Ingolia et al. 2009, PMID 19213877).

12) In Fig 1E (and also 1C), it would be very informative to have the total RNA channel plotted along with the RPF. This would help for the appreciation of changes specifically in TE. Furthermore, for genes listed in Fig S1H, it would very helpful if RPF plots were shown along with the uORF landscape cartoon.

13) Regarding Fig 2C the authors write on page 7: "Knockdown of eIF2D in DENR KO cells further reduces ATF4 protein levels and completely abolished stress-inducibility of ATF4". However, it can be observed in Fig 2C that stress-inducibility (compare KO -2D +/- tunicamycin) is not completely abolished, but greatly reduced.

14) Fig.4E and 4F: Is the difference in ASNS mRNA expression and, more importantly, protein levels between DENR-KO/eIF2D KD and DENR-KO/Rluc significant? There seems to be a trend but the significance is not clear in 4E and 4E'. Added effect of eIF2D is not really convincing in 4F.

15) The section on promotion of ATF4 by DENR-MCTS1 / eIF2D, page 12, is rather lengthy and mostly suitable for discussion. It was recently reported that high DENR expression indicated poor prognosis of cancer patients (Wang et al., #37 in MS). This publication is only cited in introduction but should be cited in relevant results and discussion sections as well.

16) In Fig S3B/C, the high sequence similarity between a-Raf uORF1 and ATF4 uORF2 between several species (mostly mammals) is reported as evolutionary conservation. Was a relevant test of conservation performed? Statistics are not reported.

17) Fig S6A: The color legend and detailed description of groups is missing.

Reviewer #2 (Remarks to the Author):

This manuscript addresses the roles of DENR and MCT1 in translational control. DENR and MCT1 were reported to contribute to ribosome reinitiation following translation of short upstream ORFs. This manuscript uses experiments involving cell culture, ribosome profiling, mRNA and protein measurements, and translational reporters, to suggest that DENR and MCT1 promote reinitiation after longer upstream ORFs with specific penultimate codons. Furthermore, the manuscript suggests that this reinitiation process features “tRNAs that remain attached to 40S ribosomes after translation termination.” This reinitiation process is critical for ATF4 translational control in the Integrated stress response (ISR), which features short 5′-proximal uORFs that provide for reinitiation that enables ribosomes to scan through an inhibitory uORF (designated #3 in this manuscript) that overlaps out-of-frame with the ATF4 coding sequence.

The processes to define reinitiation of translation is an important and timely question and this manuscript provides some compelling evidence to support the central thesis that DENR and MCT1 enhance ribosome reinitiation following translation of certain uORFs and this facilitates translation of genes, such as ATF4, which have important cancer implications. The significance of this question and broad interest warrant some enthusiasm for the manuscript. The experimental strategies and appropriate and the manuscript generally flows in a logical fashion.

However, there are major concerns. There are gaps in some key experiments. The finer details of the translation reinitiation were difficult to follow. Some of the experimental results did not appear to fully support the stated conclusions. Experimental results need to be more fully supported by quantitative descriptions and by statistical analyses. Finally, some key conclusions, such as “tRNAs that remain attached to 40S ribosomes after translation termination” (as stated in the abstract) are proposed ideas from the reporter data but not directly tested. Addressing these concerns is important to support the stated conclusions and bolster the rigor of the manuscript.

Reviewer concerns:

1. The conclusion that tRNAs that remain attached to 40S ribosomes after translation termination” is a proposal and not tested. The idea that “some aspect of tRNAs identity, perhaps its interaction with the ribosome, that determines DENR dependence” is not sufficiently tested. tRNAs were not tested. The reporters have not been systematically tested for changes in sequences versus amino acid codons (Fig

3). There is an argument for Ala codons GCG versus GCC, so one can provide a simplified argument for G flanking the stop codon; however, this idea is largely negated by the insensitivity of the Arg codon CGG (Suppl Fig 4e). Overall, there are some broad concepts here, but finer details were problematic for this reader. More conservative conclusions here appear to be warranted.

2. In Fig 2 immunoblots and others, there are examples where specific protein measurements are stated to change but by eye do not change much. For example, in Fig 2a p-eIF2alpha does not appreciably change as one would predict in response to tunicamycin. There is a high basal p-eIF2a. This is also true for Fig 2b. The text says otherwise. The key immunoblot panels should be quantitated. If one wishes to find subtle changes, one can imagine a modest increase in p-eIF2alpha in Fig. 2b lanes 7 and 8 with rescue of DENR. Furthermore, the immunoblot panels should not be narrowly cropped and should include MW markers.

3. Bar graphs should show statistical significance and key differences should be described quantitatively in text. If basal levels change (=1) this should be factored in. This is especially important for panels such as Fig. 3d where there are modest changes and it is not clear if the normalized WT changes.

4. In Fig. 3A, there should be a clear assessment of the contributions of uORF1 and uORF2 in ATF4 translational control. The text says and/or and it is not clear whether uORF1 in this reporter contributes to induction of ATF4 translational expression. For the primary ATF4 reporter used extensively here was the 5'- transcription start site defined?

5. In Fig 4,e' the ASNS protein levels do not appreciably change despite that described in the text. This is noteworthy for +/- tunicamycin and for +/- knockdown of eIF2D. Include statistical analyses and numbers of biological replicates. The prime designations were problematic for the reader (e.g. 4d and 4d').

6. In the text describing Suppl Figs 3b-c, the ATF4 and a-Raf uORFs are evolutionarily conserved. Be clear that there are no proposed similarities between ATF4 and a-RAF.

Reviewer #3 (Remarks to the Author):

In this manuscript, Bohlen et al. identify DENR, MCTS1 and eIF2D as important regulators of ATF4 and thus important players in the integrated stress response (ISR). They claim DENR and MCTS1 are necessary for translation reinitiation and induction of ATF4 in response to stress. As ISR plays an important role in cancer cells response to stresses, they indicate DENR and MCTS1 are important for cancer cell growth. Identification of new ATF4 and ISR regulators is significant. However, the current data are not sufficient to demonstrate the importance of DENR, MCTS1 and eIF2D in ISR and its relevance to the response of cancer cells to stresses.

1) Can DENR or MCTS1 KO cells tolerate ISR stress? For example, amino acid starvation and tunicamycin

induced ER stress.

2) In Fig. 2a, why p-eIF2a was not induced by tunicamycin in WT cells? This raises concerns on the conclusion drawn from this set of experiments.

3) In Fig. 4e', ASNS expression is not correlated with ATF4. While tunicamycin induces significant upregulation of ATF4 protein, ASNS protein level was not upregulated correspondingly. In addition, while the KO cells had much less ATF4, the reduction of ASNS was not dramatic.

4) The authors claim ASNS expression during amino acid starvation condition is DENR dependent. However, only KO1 had reduction of ASNS protein, but not KO2. These data weaken the author's claim.

5) It is not appropriate to use ASNS mRNA levels as a readout for ATF4 activity. In Fig. 4e' it clearly showed that ASNS level is not correlated with ATF4. This inappropriate use of ASNS as readout for ATF4 compromises their claims.

6) The impact of ATF4 on the cancer cell growth is much more dramatic than that of DENR with or without eIF2D. But the authors claim that DENR.MCTS1 also regulates Raf and Cdk4 in addition to ATF4. One would expect DENR KO has stronger effects than ATF4 KO. It is difficult for this reviewer to reconcile the phenotype with the DENR's regulation on these important cancer cell regulators.

Reviewer #1

The manuscript by Bohlen et al. analyzes the regulation of ATF4 and other oncogenes through uORFs that operate through reinitiation involving DENR-MCTS1. First, the authors identified endogenous DENR-regulated transcripts in HeLa cells by ribosome profiling (Figure 1) and observed that these carried not only short, but also long uORFs. Of note, the same authors' previously published studies (caveat: these were not based on endogenous targets but mainly on reporters) had postulated that DENR operated mainly via short and even ultrashort (1 aa) uORFs. Among the most affected endogenous targets were ATF4 (well-known for its stress-dependent uORF regulation) and some other oncogenes. In Figure 2, the authors systematically and diligently dissect the requirements of ATF4, c-Ras and a-Raf translation for uORF regulation by DENR-MCTS1; in addition, they uncover a role for eIF2D in reinitiation events. In Figure 3, the dissection goes further in detail leading to the interesting finding that for the examples tested, the ultimate/penultimate codon identity on long uORFs determines DENR-dependent reinitiation. Finally (Figure 4), the authors try to link the observed regulation to cancer. The authors present experiments on cell growth in the presence/absence of DENR, eIF2D and ATF4, and then focus mainly on one ATF4 target, asparagine synthetase (ASNS). ASNS levels show correlation with DENR expression across cancers.

This is an interesting study that contains a lot of data and that is well designed and presented, in particular for the first three-quarters (Figures 1-3). The most interesting point is the ability of certain codons at -2 and -1 positions within uORFs to render them DENR-dependent. An important task for the revision should lie in further analyzing this finding by metatranscript analyses around the uORF regions in order to determine if such codons correlate with DENR-dependent translational activity genome-wide (i.e. beyond the examples). I am a bit less impressed by the relevance of Figure 4 -- the link to ATF4 expression and cancer is somewhat weak. I believe DENR and alikes were already shown to have prognostic value in cancer in recent publications. The link with ATF4 provides some novelty, but in order to go further than simple correlations, it would be important to show, as a minimum, rescue of the growth/survival phenotype of DENR knockdown by ATF4 expression. A few other points are listed below as well.

We thank the reviewer for the positive assessment and the constructive suggestions.

We have added new data to the manuscript to address the reviewer's comments:

1) We performed two types of genome-wide analyses (80S footprinting and 40S footprinting) to support the conclusion that the -1 and -2 codons induce DENR-dependence: We first asked whether the uORFs upstream of DENR-target mORFs

(Fig. 1c) are enriched for particular penultimate codons. We analyzed all uORFs that are potentially translated in HeLa cells because they have at least one 80S ribosome footprint on their start codon (n=14892). Of these, 1220 are in the 5'UTRs of DENR targets. In this group of 1220 uORFs, four penultimate codons were enriched, including ATG^{Met} and GCG^{Ala} (Supplementary Fig. 5e). Conversely, we asked whether mORFs of transcripts containing uORFs with individual penultimate codons are up- or down-regulated upon DENR loss-of-function. We grouped mRNAs with potentially translated uORFs by uORF penultimate codon, and found that the penultimate codons CTG^{Leu}, ATG^{Met} and GCG^{Ala} were among the ones that correlated with reduced mORF translation efficiency upon DENR^{KO} genome-wide (Supplementary Fig. 5f).

We recently established 40S ribosome footprinting in human cells (<https://doi.org/10.1101/806364>), which enables the visualization and quantification of footprints from scanning 40S ribosomes in 5'UTRs and recycling 40S ribosomes on stop codons. Using this method, we asked whether genome-wide certain penultimate codons cause 40S recycling defects upon DENR loss-of-function. Indeed, as shown in new Figures 4a-f and Supplementary Figures 6c-d, loss of DENR causes a defect in 40S recycling on the stop codons of both main ORFs and uORFs (Fig. 4a-b). However, not all ORFs show a defect. For instance, the stop codon on the mORF of TUBA1B does not (Fig. 4c) whereas the stop codon of RAN does (Fig. 4d). This correlates with the identity of the ORF penultimate codon, but not other codons (Supp. Fig. 6c). Specifically, loss of DENR leads to a genome-wide defect in 40S recycling when the penultimate codon is one of 14 codons (Fig. 4e), amongst which are the ones we had previously identified from luciferase assays as causing defects in translation reinitiation in the absence of DENR.

Together, we believe these data give strong genome-wide support to the fact that the penultimate codon of an ORF is critical in determining the functional outcome of DENR loss-of-function.

2) We provide new data showing that ATF4 overexpression can partially rescue the proliferation defect of DENR^{KO} cells (Fig. 5d). Of course, DENR^{KO} cells also have defective translation of other genes that are important for cell proliferation such as a-Raf, c-Raf, and Cdk4, so one cannot expect more than a partial rescue when reconstituting only ATF4 expression. We also strengthened the correlation of ATF4 activity with DENR-MCTS1 mRNA expression in cancer by using a published expression signature for ATF4 activity (Fig. 5i).

Main points:

1) One of the novel, interesting findings of the manuscript is the importance of certain codons at penultimate and final positions of uORFs in determining DENR dependency. While the extensive testing of different codon combinations using the

dual-luciferase system elegantly demonstrates the specificity of GCG/CTG codons for the hand-picked examples, it is not clear to what extent these codons are used and can be associated with DENR-dependent regulation transcriptome-wide. Were these particular codons (or maybe others) enriched at the penultimate and ultimate positions within the transcript group regulated by DENR (e.g. the 518 down-translated genes from Fig. 1, or a similar set of genes). A metagene analysis on the landscape of all uORFs should show an enrichment of such codons correlating with down-regulation in DENR-KO.

We thank the reviewer for this important suggestion. As mentioned above, we have now done this analysis, and this is indeed the case. As shown in Suppl. Fig. 5e, the penultimate codons A-GCG, R-CGA, L-CTA and M-ATG are enriched in uORFs upstream of the 517 down-translated genes compared to all other genes in the genome. (The other penultimate codon we had identified from the luciferase assays (L-CTG) is also enriched but with a significance of $p=0.15$).

2) The proliferation defect in DENR-KO cells (Figure 4A-B) and siATF4 samples is quite striking, since ATF4 is anyway not highly expressed in cells. Given that DENR and EIF2D have hundreds of direct targets, it is not clear if the loss-of-function effect really comes from ATF4, as postulated. It is therefore important to show that the phenotype in DENR-KO/si eIF2D can be rescued by expressing additional ATF4. This is particularly important as the authors claim that the DENR phenotype could be due to low ATF4 expression. However, a direct positive test is missing.

We have now done this analysis. As shown in Fig. 5d, the proliferation defect of DENR^{KO} cells can be partially rescued by ATF4 expression.

Several points are worth mentioning:

1) This experiment was technically challenging. For reasons we do not understand, ATF4 overexpression was quickly lost upon reseeding cells from 6-well plates (for transfection) into 96-well plates for the proliferation assay. As a technical consequence, we could not perform this assay using the setup which we had previously optimized (transfecting at higher density and then reseeding for proliferation curves). We then spent a long time optimizing a transfection protocol directly in 96-well plates, which was challenging because a cell density that was too low led to cell death upon transfection, whereas a higher cell density (usually used for transfections) caused the wells to become confluent after 2 days, so the proliferation assay could not be done. Finally, we reached a condition where transfection efficiency was still good but cells were only mildly stressed. This is why the proliferation curve in Fig. 5d show a lag phase before they start proliferating exponentially.

2) A partial rescue is the most we can expect from this experiment. DENR^{KO} cells also have defective translation of other genes that are important for cell proliferation such as a-Raf, c-Raf, and Cdk4, so one cannot expect more than a partial rescue when reconstituting only ATF4 expression. For this reason, we had written in the original manuscript

“translation of additional DENR target genes including Cdk4, a-Raf, and c-Raf, likely also contribute to the DENR phenotype.”

3) It is worth mentioning that ATF4 levels have been found to influence cancer cell proliferation also in a number of other studies (PMID 22102693, 28553953, 20473272). It appears that the low levels of ATF4 present in cancer cells are physiologically important, since no exogenous stress was required in these studies for ATF4 to effect proliferation.

3) First point regarding the 1-aa uORFs: There is in general a bit of overexcitement about the finding that not only 1-aa long uORFs but longer ones are involved in DENR dependent regulation as well, as if this is an unexpected and very new finding. As far as this reviewer understands, the 1-aa finding stemmed largely from reporter assays from the lab's previous studies. Other recent papers did not confirm the “short uORF” hypothesis (e.g. PMID 30982898). All this is not presented or discussed in a sufficiently balanced fashion. The novelty of the finding should thus be toned down, and the interpretation and reporting should be adapted accordingly.

We have rephrased the text to remove excitement. We did not wish to present the literature in an unbalanced fashion. We simply need to point out that most longer uORFs are not DENR dependent, and only a specific subclass of longer uORFs is. To our knowledge, this has not been reported before and is therefore novel.

We feel it is not really correct, as the Reviewer writes, that “other recent papers did not confirm the ‘short uORF’ hypothesis (e.g. PMID 30982898)”. This paper simply did not investigate this issue, because they excluded 1-aa uORFs from all analyses, as they write:

“uORFs were annotated and considered as translated with the following criteria: (i) started with AUG, CUG, GUG or UUG, (ii) had an in-frame stop codon within the 5' UTR or overlapping/within the CDS, (iii) were at least 9 nt long (including stop codon), (iv) had a coverage >25%, (v) showed a frame preference and (vi) the preferred frame was the one corresponding to the uORF 5' end/start codon.”

4) Second point regarding 1-aa uORFs: In Fig 1C, the authors describe that Droscha displayed increased ribosome density on the

1-aa uORFs. However, this is very difficult to observe in this Figure due to several reasons: 5-nt windowing creating a diffused signal distribution, insufficient zoom on the X-axis and close proximity of 1-aa uORFs and longer ones to each other. Furthermore, in Fig 1A, the 5'-end of the reads were used for mapping (not A-, or P-site corrected positions) which shifts the signal approximately 15 nt upstream. However, in 1C, some of the strongest peaks can be observed downstream of the 1-aa uORF sites. To properly distinguish if the ribosome footprint densities are overlapping specifically with the 1-aa uORF sites a more precise computational approach at least at codon resolution is necessary.

We thank the reviewer for catching this problem. The issue was an error in the assembly of the figure: The transcript diagram was shifted a bit to the left, hence the start of the transcript did not align with 0 on the graph x-axis anymore (Reviewer Figure 1).

This is now fixed. A detailed view is in Reviewer Figure 2 (please note the -15 nt offset):

Reviewer Figure 3 shows a detailed P-site analysis for the DROSHA mRNA around the uORF 203-208. A clear increase in ribosomes with the P-site positioned on the

ATG codon of the stuORF can be observed in DENR^{KO}, while no increase in footprint density upon DENR^{KO} is observed on the ATG of the longer uORF at position 220.

5) Third point regarding 1-aa uORFs: In Fig 1D, it was reported that transcripts containing one or more 1-aa uORF were excluded from all other uORF groups. However, it is not clear if the 1-aa uORF group contained also longer uORFs. Also, size (number of transcripts) in each group should be reported. How were genes with multiple transcript models handled? Was same uORF used in different groups? A negative correlation between uORF size and number of non-overlapping uORFs is expected; was this accounted for?

The reviewer raises a number of important points. We respond to each point one-by-one:

- However, it is not clear if the 1-aa uORF group contained also longer uORFs-

Yes – we have now made this clear in the figure legend and the Methods.

Transcripts in each group can also contain uORFs from the other groups, except transcripts with 1-aa uORFs, which we excluded from all other groups (and of course transcripts with no uORFs). The point we want to make in this figure is that the drop in TE seen in the categories with longer uORFs is not due to the presence of 1-aa uORFs.

We already know that a 1-aa uORF *per se* can induce DENR-dependent reinitiation, hence in the original figure we did not remove longer uORFs from this category. We

now also include as Supplemental Fig. 2f an equivalent graph showing that transcripts containing only 1-aa uORFs (i.e. not longer uORFs) also show a highly significant reduction in translation efficiency upon loss of DENR, as expected from our previous publications.

-Also, size (number of transcripts) in each group should be reported.-

We have added this to Figure 1e and Suppl. Fig 2f.

-How were genes with multiple transcript models handled?-

We have now added a paragraph to the methods explaining the analysis for this figure. Since 80S reads are in the ORFs, and usually transcript isoforms with different 5'UTRs share a common ORF, it is not possible to allocate 80S reads to the correct 5'UTR isoform. (Even with RNA-seq data to exclude non-expressed isoforms, as long as there are 2 expressed isoforms for a gene, we would not know which 80S footprint in the ORF corresponds to which isoform.) Hence two approaches are possible. One option is to only analyze genes with single transcript isoforms. These, however, are very few and hence the analysis is noisy. We took the alternative approach and allowed multiple mapping of reads to all transcript isoforms. This means there will be both 'false-positives' and 'false-negatives' (ie transcript isoforms that looks like they are changing in TE although actually the reads are coming from a different isoform, as well as transcript isoforms that look like they are not changing in TE, even though they are, but there is a more highly-expressed isoform that does not change in TE, and the reads should actually map to this one). However, over the whole transcriptome, these effects will create some noise but we still see significant signal emerging.

-Was same uORF used in different groups?-

Yes, the groups are not mutually exclusive. They contain all transcripts that harbour the indicated feature, except for transcripts with 1AA uORFs which were excluded from all other groups.

-A negative correlation between uORF size and number of non-overlapping uORFs is expected; was this accounted for?-

We are not sure we understand this point. By "overlapping uORFs" we meant uORFs that overlap the main ORF start codon. Since these are actually not relevant here, as reinitiation cannot occur on them, we have now removed this group from the figure.

6) This manuscript adds to a number of studies that identify DENR targets – e.g. Schleich et al. 2014 and 2017 (the authors' lab) and Castelo-Szekely et al. 2019 (PMID 30982898). The latter also suggested ATF4/5 as targets and identified several cancer-related genes (Klhdc8a, Map2k5). Can the authors discuss determine and discuss the overlap between the studies (e.g. was Atf4 also a target in Drosophila; is Drosha a target across species/systems?).

We now include this analysis in Suppl. Figure 2c. Castelo-Szekely et al. 2019 (PMID 30982898) was done in mouse cells. Hence we determined the human homologs of these targets and checked for overlap of this set of genes to our 517 identified DENR targets. Out of 212 mouse DENR targets, 203 have human homologs. Of these, 47 overlap with our set. The likelihood of this happening by chance is 10^{-14} (calculated by binomial distribution, using as a background set of 10529 detected mouse genes of which 9663 have known human homologs). Note that the mouse NIH3T3 cells used in PMID 30982898 and the HeLa cells we used here have different gene expression landscapes, hence many genes expressed in one cell line are not expressed in the other, which can account for the non-overlapping genes.

Our manuscript is a back-to-back co-submission with a manuscript from Hyung Don Ryoo's laboratory where they discover that Drosophila ATF4 is a target of DENR•MCTS1/Ligatin. In their manuscript they start by generating an *in vivo* fluorescent reporter for ATF4 activity in Drosophila and they combine this with an RNAi screen. In this way, they also discover that DENR-MCTS1 and Ligatin are required for ATF4 translation *in vivo* in the fly. Hence this is conserved to Drosophila. The two studies complement each other nicely because we start with DENR and arrive at ATF4 while the Ryoo study starts with ATF4 and arrives at DENR. Furthermore, we have a mechanistic focus in our manuscript whereas the Ryoo manuscript has an *in vivo* physiological/functional focus. We have added a reference to their manuscript in our revised manuscript.

7) Fig. 1B it is not immediately clear from which values the Z-scores were calculated – change in log-transformed TE or simply change in TE?

We thank the reviewer for catching this lack of clarity. We are calculating the z-scores from log2 transformed TE and have updated the labelling in the figure (now Figure 1C).

Was a test of significance performed which is well adapted for TEs (e.g. Xtail)? How many of these 518 were significantly down-translated in DENR-KO?

We have now performed the Xtail analysis (excluding transcripts with less than 64 reads per transcript). This arrives at similar results (Reviewer Figure 4, and new Supplemental Figure 2b). With a cutoff of $p < 0.05$, it identifies 38 regulated genes, of which 37 are down-regulated and 1 is up-regulated. The down-regulated genes include ATF4 as a top hit, Drosha, PIK3R2 and MAP2K6. Two hits which we validate experimentally in our manuscript, ARAF and Raf1, do not pass this threshold in the Xtail analysis, hence the X-tail analysis is likely a bit too stringent.

8) In general, annotation of uORFs is not sufficiently communicated. Were ribosome profiling data utilized to identify translated uORFs? Were non-canonical start codons allowed, or just ATG, in identification of putative uORF positions?

We have now added to the Methods a description of uORF annotation.

The uORFs shown in the schematic diagrams of the transcripts (e.g. Figures 1d, 1f, Supplemental Fig. 2e, 2h, 2i) all start with ATG. The ribosome profiling traces are provided in these figure panels alongside the schematic diagrams.

The uORFs of ATF4, a-Raf and c-Raf all start with ATG and are extensively functionally characterized in Figs 2 and 3.

In Figure 1e, the uORFs all start with ATG. Ribosome footprinting was not used to determine which uORFs are translated. We believe this is a reasonable approach because we are more worried about false negatives than false positives. uORFs that harbour regulatory activity can easily be missed in footprinting analyses (e.g. Ribotaper) if they are not strongly translated, or if they are short (1-3 a.a.) because they lack obvious triplet periodicity which is detected by these analysis pipelines. False positives are less of a concern since some transcript that is falsely classified as having a translated uORF would, if anything, make the effect of that group that it is attributed to less strong. Despite this we still observe highly significant effects.

For the new analyses presented in Figure 4 and Supplemental Figures 5 & 6, we introduce a classification for 'potentially translated' uORF, where we count 80S ribosome footprints on the uORF start codon (10 nt window). With this classifier, we find 14892 uORFs to be translated in our dataset. Here too, we set the threshold for identifying a uORF as being 'potentially translated' purposefully low, to avoid false-negatives. False-positives will only dilute the signal and statistical enrichments that we are showing in these figures.

9) Fig S4A #9 does not agree with Fig 3C #4. In Fig 3C, Codon-1 GCG>TCC mutation seems to have lost DENR-dependence while in Fig S4A, the same mutation is clearly DENR dependent. Does it imply variability within the assay, or certain codons? Can the authors add more biological replicates and discussion to clarify such cases?

We had split the figures relating to a-Raf uORF2 between the main and Supplemental Figures because the assays for the reporters shown had been carried out on separate days. As the reviewer points out, this led to some biological variability. As suggested, we have now performed more biological replicates of these experiments, making sure to include all the relevant reporters together in one experiment to be assayed in parallel, and present these data in Figure 3c. From these biological replicates we conclude that mutating only codon -1 removes most of the DENR dependence, but it is not sufficient to remove all DENR dependence. For this also codon -2 needs to be mutated.

10) Some Western Blots lack quantification (for example Fig 4G). Are the quantifications already normalized to loading control?

Yes – quantifications are normalized to loading controls in all cases. We have now included a statement to this effect in the “Data analysis and statistics” section of the Materials & Methods.

We have now added quantifications to Fig 5g (previously 4g) and some other blots. (We did not put quantifications where the magnitude change is large enough to be visually obvious.)

Other comments:

11) Intro + abstract: The 1000-fold change in TE has not been reported in any empirical study. The Schwannhausser paper that is cited infers translation from a model. The authors may want to use a different reference that actually shows factor 1000,

or put this value down to more common estimates that are rather 10-fold (e.g. ES-cell data, Ingolia et al. 2011, PMID 22056041) to 100-fold (e.g. budding yeast Ingolia et al. 2009, PMID 19213877).

We thank the reviewer for pointing this out, and have reworded the abstract and intro to read “varies considerably” instead of putting a hard number on it.

12) In Fig 1E (and also 1C), it would be very informative to have the total RNA channel plotted along with the RPF. This would help for the appreciation of changes specifically in TE. Furthermore, for genes listed in Fig S1H, it would very helpful if RPF plots were shown along with the uORF landscape cartoon.

The curves that are shown are already normalized to total RNA-levels (one value for the entire mRNA), and we have now pointed this out more clearly in the figure legend. We find this graphically easier to interpret than showing three additional RNA-seq curves in the panel (one per genotype) and requiring readers to visually normalize. (If the reviewer is concerned about a position-specific sequencing bias, this should be the same in both WT and DENR^{KO} cell samples, hence it normalizes away when comparing TE between the two.)

As requested, we have added similar ribosome density plots for additional DENR targets in Supplemental Figure 2h, k, and l.

13) Regarding Fig 2C the authors write on page 7: “Knockdown of eIF2D in DENR KO cells further reduces ATF4 protein levels and completely abolished stress-inducibility of ATF4”. However, it can be observed in Fig 2C that stress-inducibility (compare KO -2D -/+ tunicamycin) is not completely abolished, but greatly reduced.

We agree and have rephrased the interpretation of this blot in the results section:

“Knockdown of eIF2D in the DENR^{KO} cells further reduced ATF4 protein levels and strongly impaired the stress-inducibility of ATF4 (Fig. 2c).”

14) Fig.4E and 4F: Is the difference in ASNS mRNA expression and, more importantly, protein levels between DENR-KO/eIF2D KD and DENR-KO/Rluc significant? There seems to be a trend but the significance is not clear in 4E and 4E'. Added effect of eIF2D is not really convincing in 4F.

We have toned down the language in how we interpret these data,

“Induction of ATF4 with tunicamycin leads to elevated ASNS transcription (Fig. 5e) and mildly elevated ASNS protein (lanes 1 & 7, Fig. 5f). ASNS

induction is reduced in DENR knockout cells at the mRNA (Fig. 5e), and protein levels (lane 9 Fig. 5f).”

Additionally, changes in Figure 4E (now Supplemental Figure 7d) have been statistically assessed and significance is indicated. (Both the contribution of DENR and of eIF2D to ATF4 transcriptional activity (ie ASNS levels) are statistically significant). Please note that the ASNS mRNA levels are most important as a readout for ATF4 activity, since ATF4 regulates ASNS transcriptionally. The changes in ASNS proteins levels are smaller in magnitude compared to the changes in mRNA levels because tunicamycin also causes a global drop in mRNA translation.

15) The section on promotion of ATF4 by DENR-MCTS1 / eIF2D, page 12, is rather lengthy and mostly suitable for discussion. It was recently reported that high DENR expression indicated poor prognosis of cancer patients (Wang et al., #37 in MS). This publication is only cited in introduction but should be cited in relevant results and discussion sections as well.

We have added a citation of Wang et al to the Discussion. (We did not find a sentence in the current version of the Results section where this citation would fit.)

To address the reviewer’s concern regarding the length of page 12, we have tightened the wording in this section so that each concept is introduced with max. 2 sentences. (We feel it is more reader-friendly to briefly introduce the relevant concepts at the spots they are needed – in the Discussion they would be more disconnected from the data.)

16) In Fig S3B/C, the high sequence similarity between a-Raf uORF1 and ATF4 uORF2 between several species (mostly mammals) is reported as evolutionary conservation. Was a relevant test of conservation performed? Statistics are not reported.

We have rephrased this section because we simply wanted to point out that a similar sequence is present in different animal species.

17) Fig S6A: The color legend and detailed description of groups is missing.

Thank you for catching this mistake. We have added a color legend to the figure and a description of the groups to the figure legend.

Reviewer #2

This manuscript addresses the roles of DENR and MCT1 in translational control. DENR and MCT1 were reported to contribute to ribosome reinitiation following translation of short upstream ORFs. This manuscript uses experiments involving cell culture, ribosome profiling, mRNA and protein measurements, and translational reporters, to suggest that DENR and MCTS1 promote reinitiation after longer upstream ORFs with specific penultimate codons. Furthermore, the manuscript suggests that this reinitiation process features "tRNAs that remain attached to 40S ribosomes after translation termination." This reinitiation process is critical for ATF4 translational control in the Integrated stress response (ISR), which features short 5'-proximal uORFs that provide for reinitiation that enables ribosomes to scan through an inhibitory uORF (designated #3 in this manuscript) that overlaps out-of-frame with the ATF4 coding sequence.

The processes to define reinitiation of translation is an important and timely question and this manuscript provides some compelling evidence to support the central thesis that DENR and MCTS1 enhance ribosome reinitiation following translation of certain uORFs and this facilitates translation of genes, such as ATF4, which have important cancer implications. The significance of this question and broad interest warrant some enthusiasm for the manuscript. The experimental strategies and appropriate and the manuscript generally flows in a logical fashion.

However, there are major concerns. There are gaps in some key experiments. The finer details of the translation reinitiation were difficult to follow. Some of the experimental results did not appear to fully support the stated conclusions. Experimental results need to be more fully supported by quantitative descriptions and by statistical analyses. Finally, some key conclusions, such as "tRNAs that remain attached to 40S ribosomes after translation termination" (as stated in the abstract) are proposed ideas from the reporter data but not directly tested. Addressing these concerns is important to support the stated conclusions and bolster the rigor of the manuscript.

We thank the reviewer for the positive assessment and the constructive suggestions. We have now incorporated the reviewer's feedback and substantial amounts of new data to make the manuscript stronger.

Reviewer concerns:

1. The conclusion that tRNAs that remain attached to 40S ribosomes after translation termination" is a proposal and not tested. The idea that "some aspect of tRNAs identity, perhaps its interaction with the ribosome, that determines DENR dependence" is not sufficiently tested. tRNAs were not tested. The reporters have not been systematically tested for changes in sequences versus amino acid codons (Fig 3). There is an argument for Ala codons GCG versus GCC, so one can provide a simplified argument for G flanking the stop codon; however, this idea is largely negated by the insensitivity of the Arg codon CGG (Supplemental Fig 4e). Overall, there are some broad concepts here, but finer details were problematic for this reader. More conservative conclusions here appear to be in warranted.

We have done two things to address this concern. Firstly, as detailed below, we have added substantial amounts of new data that support the conclusion that the identity of the penultimate codon in the ORF, and not the encoded amino acid, determines the functional outcome of DENR loss-of-function. Admittedly, we do not distinguish between the role of the codon itself, and the role of the tRNA that decodes it. Therefore, secondly, we have rephrased the text to focus on the codon and make very clear that the contribution of the tRNA is only a model. We deleted the sentence cited by the reviewer, and now explicitly write:

"Our data do not allow us to distinguish between the role of the penultimate codon itself, and the role of the tRNA that decodes it."

1) New Data:

We have added two entirely new figures (Main Fig. 4 and Supplemental Figure 6) where we test post-termination 40S recycling genome-wide using 40S ribosome footprinting. We recently established 40S ribosome footprinting in human cells (<https://doi.org/10.1101/806364>), which enables the visualization and quantification of footprints from scanning 40S ribosomes in 5'UTRs and recycling 40S ribosomes on stop codons. Using this method, we asked whether certain penultimate codons cause 40S recycling defects genome-wide upon DENR loss-of-function. Indeed, as shown in Figures 4a-f and Supplementary Figures 6c-d, loss of DENR causes a defect in 40S recycling on the stop codons of both main ORFs and uORFs (Fig. 4a-b). However, not all ORFs show a defect. For instance, the stop codon on the mORF of TUBA1B does not (Fig. 4c) whereas the stop codon of RAN does (Fig. 4d). This correlates with the identity of the ORF penultimate codon, but not other codons (Supp. Fig. 6c). Specifically, loss of DENR leads to a genome-wide defect in 40S recycling when the penultimate codon is one of 14 codons (Fig. 4e), amongst which are the ones we had previously identified from luciferase assays as causing defects in translation reinitiation in the absence of DENR (L-CTG, A-GCG and M-ATG). This agrees with our luciferase assays and shows that the identity of the penultimate codon of an ORF determines the functional outcome of loss of DENR.

As the reviewer correctly points out, there appears to be no simple rule that can be formulated from the pattern of enriched versus de-enriched codons. We agree that it cannot be a terminal G, since also L-CTT and L-CTC are enriched in Figure 4e, in addition to the luciferase reporter data cited by the reviewer.

Additionally, we have assessed whether exchanging the A-GCG codon with A-GCC can abrogate the DENR-dependence in the original aRaf uORF1 and ATF4 uORF2, and indeed this is the case (Figure S5 a + d), showing that the encoded amino acid cannot determine DENR-dependence.

2) Text changes

There are several reasons to think that the penultimate tRNA, and not the penultimate codon, determines the outcome of DENR loss-of-function, not least of which is the fact that DENR has been shown to act on the tRNA to evict it from post-termination 40S ribosomes. The genome-wide 40S footprinting data we present above show that indeed this recycling defect observed upon loss of DENR depends on the penultimate codon, and hence also the tRNA that is in the P site. Further evidence for the idea that the tRNA identity plays a role here can be found in Skakbin et al. 2013 Molecular Cell, where the Pestova group has shown that some tRNAs do not dissociate as readily from terminating 40S ribosomes as others. In Figure 1J they show that the C-TGC tRNA spontaneously dissociates from the terminating 40S ribosome but the L-CTT tRNA does not. Note that in our analysis of main ORF stop codons, penultimate L-CTT is strongly enriched in the set of genes that have a 40S peak upon DENR KO, while C-TGC tRNA is strongly de-enriched. Furthermore, they show in Figure 5 C and G that either eIF1 + eIF1A or Ligatin can actively dislocate this L-CTT tRNA from the terminating 40S ribosome.

We have reworded the text, however, to focus on the codon, and we explicitly write that we cannot distinguish between the codon and the tRNA.

2. In Fig 2 immunoblots and others, there are examples where specific protein measurements are stated to change but by eye do not change much. For example, in Fig 2a p-eIF2alpha does not appreciably change as one would predict in response to tunicamycin. There is a high basal p-eIF2a. This is also true for Fig 2b. The text says otherwise. The key immunoblot panels should be quantitated. If one wishes to find subtle changes, one can imagine a modest increase in p-eIF2alpha in Fig. 2b lanes 7 and 8 with rescue of DENR. Furthermore, the immunoblot panels should not be narrowly cropped and should include MW markers.

We are somewhat perplexed by the reviewer's comment 'The text says otherwise'. We do not think we claimed anywhere in the original manuscript that eIF2 α

phosphorylation changes on any of our blots (including Fig 2) ? We searched again the manuscript we originally submitted, and did not find this anywhere ?

That said, the reviewer is correct in pointing out that the p-eIF2 α blots have high basal p-eIF2 α . Indeed, eIF2 α phosphorylation is notoriously tricky to detect, and that is also the case here. We know the tunicamycin treatment is working and inducing the Integrated Stress Response (ISR): We include new data showing that tunicamycin causes a strong inhibition of global protein translation, visible on a polysome profile (Suppl. Fig. 3b). Furthermore, it strongly induces ATF4 protein levels (original Figure 2a lanes 1-2), which would not be the case if it did not induce the ISR. So this is a technical issue with the p-eIF2 α antibody. In principle, two options are possible. Either it has a high background in the control condition, or it does not detect well the increase in eIF2 α phosphorylation caused by tunicamycin. We now analyze this in Suppl. Figure 3a. We treated the samples with lambda phosphatase to dephosphorylate them. As a positive control for the phosphatase treatment, we see that S6K phosphorylation is almost completely removed. In contrast, the p-eIF2 α signal detected with two different commercially available antibodies (from Cell Signaling and Abcam) does not drop (compare lanes 1 vs 4). Hence this means that the p-eIF2 α signal detected in the control condition is due to the antibodies detecting non-phosphorylated protein. (To some extent, this is the case for all phospho-antibodies, and the stoichiometry of phosphorylation is important in determining the signal/noise ratio. In this case, it is likely that HeLa cells have a good amount of eIF2 α protein, and little of it is phosphorylated in the control condition). We also treated cells with arsenite prior to lysis, and this strongly induces the ISR and gave a clear p-eIF2 α signal using the same detection conditions (lane 3). Hence, this means our tunicamycin treatment conditions induce a more mild ISR, nonetheless, it is sufficient for our purposes because the induction of ATF4 is clear.

As requested, we have added quantifications to all main western blots. If the reviewer has specific panels or bands in mind where additional quantification should be added we are happy to do so.

Uncropped versions of all blots, with the size markers, are now provided in the Source Data File (If we crop each WB panel less, they do not fit in the figures anymore).

3. Bar graphs should show statistical significance and key differences should be described quantitatively in text. If basal levels change (=1) this should be factored in. This is especially important for panels such as Fig. 3d where there are modest changes and it is not clear if the normalized WT changes.

We have now added statistical tests to all luciferase reporter bar graphs indicating the degree of significance, including specifically also Figure 3D.

In Fig. 3D the black bars are not all exactly equally well translated in WT cells (also no major differences), but it is important to normalize this out to look at DENR-dependent effects. Reviewer Figure 5 shows what Figure 3D looks like when the black bars are not normalized to 1.

Reviewer Figure 5:
Non-normalized view
of Figure 3d.

As requested, we have also added to the text quantitative statements about how much reporter expression changes upon DENR^{KO}.

4. In Fig. 3A, there should be a clear assessment of the contributions of uORF1 and uORF2 in ATF4 translational control. The text says and/or and it is not clear whether uORF1 in this reporter contributes to induction of ATF4 translational expression. For the primary ATF4 reporter used extensively here was the 5'- transcription start site defined?

We have clarified in the text that both uORFs contribute, and by how much each one drops upon DENR^{KO} (60% and 50% for uORFs 1 and 2 respectively).

We have also added exact information regarding the ATF4 5'UTR full-length reporter, including the exact transcription start site, to the Materials & Methods.

5. In Fig 4,e' the ASNS protein levels do not appreciably change despite that described in the text. This is noteworthy for +/- tunicamycin and for +/-knockdown of eIF2D. Include statistical analyses and numbers of biological replicates. The prime designations were problematic for the reader (e.g. 4d and 4d').

We have renamed the figure panels to remove primes in the figure panel labels.

We agree that ASNS protein levels do not change as strongly as the mRNA levels. This is probably because tunicamycin causes a global drop in translation, hence the transcriptional changes are blunted at the protein level. We mention this clearly in the text now. That said, the mRNA is the relevant parameter for ATF4 activity because ATF4 is transcriptionally regulating ASNS.

As requested, we have added statistical analysis to Fig 4e (now Supplemental Figure 7d) showing that both the contribution of DENR and of eIF2D to ATF4 transcriptional activity (ie ASNS mRNA levels) are statistically significant. We have added quantifications to Fig. 4e' (now Supplemental Figure 7e) showing that ASNS protein levels drop upon DENR loss-of-function. (Please note the quantifications are already normalized for protein loading). We have included information about biological replicates to the figure legend.

6. In the text describing Suppl Figs 3b-c, the ATF4 and a-Raf uORFs are evolutionarily conserved. Be clear that there are no proposed similarities between ATF4 and a-RAF.

Thanks for catching this possible misunderstanding. We now rephrased the sentence to make this clear.

Reviewer #3

In this manuscript, Bohlen et al. identify DENR, MCTS1 and eIF2D as important regulators of ATF4 and thus important players in the integrated stress response (ISR). They claim DENR and MCTS1 are necessary for translation reinitiation and induction of ATF4 in response to stress. As ISR plays important role in cancer cells response to stresses, they indicate DENR and MCTS1 are important for cancer cell growth. Identification of new ATF4 and ISR regulators is significant. However, the current data are not sufficient to demonstrate the importance of DENR, MCTS1 and eIF2D in ISR and its relevance to the response of cancer cells to stresses.

1) Can DENR or MCTS1 KO cells tolerate ISR stress? For example, amino acid starvation and tunicamycin induced ER stress.

Our manuscript is a back-to-back co-submission with a manuscript from Hyung Don Ryoo's laboratory. (We do not know if the Reviewer is also a reviewer on the Ryoo manuscript.) In their manuscript, Ryoo and colleagues also discover that DENR-MCTS1 and eIF2D are required for ATF4 translation, starting from a fluorescent reporter for ATF4 activity in vivo in *Drosophila*, and searching via an RNAi screen for factors required for ATF4 translation. The two manuscripts complement each other, because our manuscript describes experiments in human cell culture and is more mechanistic. The Ryoo manuscript describes experiments *in vivo* in *Drosophila* and focuses less on mechanism and more on the physiological relevance of this regulation. Their results show that indeed DENR•MCTS1 and eIF2D are part of the ISR. We believe their model system is much better suited to address these issues than our cells in culture.

That said, we now provide new data showing that the reduced proliferation of DENR^{KO} cells is partly due to reduced ATF4 levels, because it can be partially rescued with ATF4 expression (Fig. 5d). Of course, DENR^{KO} cells also have defective translation of other genes that are important for cell proliferation such as a-Raf, c-Raf, and Cdk4, so one cannot expect more than a partial rescue when reconstituting only ATF4 expression. Nonetheless, this shows that part of the phenotype of DENR^{KO} cells is due to impaired ATF4 and hence impaired ISR.

DENR^{KO} cells are indeed more susceptible to stress and Reviewer Figure 6 shows propidium iodide staining (assayed by FACS) as a readout for cell death after a 2 hour sodium arsenite challenge and recovery for 24 hours. This shows that indeed DENR^{KO} cells have impaired tolerance to this stress. We prefer not to show these data in our manuscript because they are important for another project in our group investigating the upstream regulation of DENR by stress.

2) In Fig. 2a, why p-eIF2 α was not induced by tunicamycin in WT cells? This raises concerns on the conclusion drawn from this set of experiments.

eIF2 α phosphorylation is notoriously tricky to detect. We know the tunicamycin treatment is working and inducing the Integrated Stress Response (ISR): We include new data showing that tunicamycin causes a strong inhibition of global protein translation, visible on a polysome profile (Suppl. Fig. 3b). Furthermore, it strongly induces ATF4 protein levels (original Figure 2a lanes 1-2), which would not be the case if it did not induce the ISR. So these data indicate there is not a concern regarding the conclusions drawn from these experiments, but rather a technical issue with the p-eIF2 α antibody. In principle, two options are possible. Either it has a high background in the control condition, or it does not detect well the increase in eIF2 α phosphorylation caused by tunicamycin. We now analyze this in Suppl. Figure 3a. We treated the samples with lambda phosphatase to dephosphorylate them. As a positive control for the phosphatase treatment, we see that S6K phosphorylation is almost completely removed. In contrast, the p-eIF2 α signal detected with two different commercially available antibodies (from Cell Signaling and Abcam) does not drop (compare lanes 1 vs 4). Hence this means that the p-eIF2 α signal detected in the control condition is due to the antibodies detecting non-phosphorylated protein. (To some extent, this is the case for all phospho-antibodies, and the stoichiometry of phosphorylation is important in determining the signal/noise ratio. In this case, it is likely that HeLa cells have a good amount of eIF2 α protein, and little of it is phosphorylated in the control condition). We also treated cells with arsenite prior to lysis, and this strongly induces the ISR and gave a clear p-eIF2 α signal using the same detection conditions (lane 3). Hence, this means our tunicamycin treatment conditions induce a more mild ISR, nonetheless, it is sufficient for our purposes because the induction of ATF4 is clear.

3) In Fig. 4e', ASNS expression is not correlated with ATF4. While tunicamycin induces significant upregulation of ATF4 protein, ASNS protein level was not upregulated correspondingly. In addition, while the KO cells had much less ATF4, the reduction of ASNS was not dramatic.

We agree that ASNS protein levels do not increase as strongly as the mRNA levels upon tunicamycin treatment. This is probably because tunicamycin causes a global drop in translation (Suppl. Fig. 3b), hence the transcriptional changes are blunted at the protein level. We mention this clearly in the text now. That said, the mRNA is the relevant parameter for ATF4 activity because ATF4 is transcriptionally regulating ASNS, and ASNS mRNA levels are clearly DENR dependent (Fig. 5e) and eIF2D dependent (Suppl. Fig. 7d).

4) The authors claim ASNS expression during amino acid starvation condition is DENR dependent. However, only KO1 had reduction of ASNS protein, but not KO2. These data weaken the author's claim.

We do see a 40%-60% drop in ASNS protein upon a.a. stress, when both DENR and eIF2D are depleted (compare lanes 7 to 10 and 12 in Suppl. Fig. 7f). This is now easier to assess since we have added quantifications to the figure. It is true that with only DENR depletion, the changes in ASNS protein levels are more mild, but this simply reflects functional redundancy between these two proteins.

5) It is not appropriate to use ASNS mRNA levels as a readout for ATF4 activity. In Fig. 4e' it clearly showed that ASNS level is not correlated with ATF4. This inappropriate use of ASNS as readout for ATF4 compromises their claims.

There seems to be a misunderstanding by the Reviewer here. Figure 4e' (now Suppl. Fig. 7e) shows ASNS protein levels, not mRNA levels. Because of the issues discussed above in response to point # 3, the magnitude in change in ASNS mRNA levels does not translate into equally large changes in ASNS protein levels because of the effect of tunicamycin on global translation. ASNS mRNA levels, however, do correlate with ATF4 protein levels. This makes sense because ASNS is a well-established, direct transcriptional target of ATF4. In PMID 15385533 it was shown that ASNS transcription is ATF4 dependent using a transcriptional reporter, and that ATF4 binds the ASNS promoter region via Ch-IP. Similar results were reported in PMID 29316436. Many more studies can be cited to prove this point. Additionally, ASNS mRNA levels decrease dramatically upon ATF4 knockdown in tunicamycin treated HeLa cells (Reviewer Figure 7). Hence we do believe ASNS mRNA levels are an appropriate readout for ATF4 activity.

Additionally, we now also use a published transcriptional signature for ATF4 activity (PMID 31263264) made of 11 genes. Using this signature, we obtain an even better correlation between DENR ($r > 0.39$) or DENR+MCTS1 expression ($r > 0.42$) and ATF4 activity across the entire TCGA pan-cancer dataset. These data are now shown in Figure 5i. Therefore, we can conclude that indeed there is a solid correlation between DENR-MCTS1 mRNA expression and ATF4 activity / ATF4 target expression across cancer samples.

6) The impact of ATF4 on the cancer cell growth is much more dramatic than that of DENR with or without eIF2D. But the authors claim that DENR.MCTS1 also regulates Raf and Cdk4 in addition to ATF4. One would expect DENR KO has stronger effects than ATF4 KO. It is difficult for this reviewer to reconcile the phenotype with the DENR's regulation on these important cancer cell regulators.

This is because the decrease in ATF4 levels in DENR^{KO} cells (Figs. 2a + b) are more mild than the decrease in ATF4 upon direct ATF4 knockdown (Suppl. Fig. 7b). Hence it makes sense that the phenotype is also more mild.

As mentioned above, we now also provide new data showing that the reduced proliferation of DENR^{KO} cells is partly due to reduced ATF4 levels, because it can be partially rescued with ATF4 expression (Fig. 5d). The partial rescue when reconstituting only ATF4 expression is in line with the fact that DENR^{KO} cells also have defective translation of other genes that are important for cell proliferation such as a-Raf, c-Raf, and Cdk4. (Please note that also in the case of a-Raf, c-Raf and Cdk4, the protein levels are reduced, but not completely gone in DENR^{KO} cells.)

REVIEWERS' COMMENTS:

Reviewer #1 (Remarks to the Author):

The manuscript has been improved and all my previous comments have been addressed. I support publication.

Of note, I would particularly like to thank the authors for the careful and detailed responses they gave in the rebuttal - I appreciate their effort.

This is interesting and important work, well described, and effectively presented.

Reviewed by David Gatfield

Reviewer #2 (Remarks to the Author):

This revised manuscript addresses the roles of DENR and MCT1 in translational control. DENR and MCT1 were reported to contribute to ribosome reinitiation following translation of short upstream ORFs. This manuscript uses experiments involving cell culture, ribosome profiling, mRNA and protein measurements, and translational reporters, to suggest that DENR and MCTS1 promote reinitiation after longer upstream ORFs with specific penultimate codons. The manuscript indicates that the roles of penultimate codons in uORFs are critical for DENR and MCTS1 to promote ribosome reinitiation following translation of regulatory uORFs. This reinitiation process is critical for ATF4 translational control in the Integrated stress response (ISR), which features short 5'-proximal uORFs that provide for reinitiation that enables ribosomes to scan through an inhibitory uORF that overlaps out-of-frame with the ATF4 coding sequence. The processes to define reinitiation of translation is an important and timely question and this manuscript provides some compelling evidence to support the central thesis that DENR and MCTS1 enhance ribosome reinitiation following translation of certain uORFs and this facilitates translation of genes, such as ATF4, which have important cancer implications. The significance of this question and broad interest warrant enthusiasm for the manuscript. The experimental strategies are appropriate and the manuscript flows in a logical fashion. Prior reviewer concerns were sufficiently addressed.

Reviewer #3 (Remarks to the Author):

The revised version has addressed most of my questions.